# Deep-water inflow event increases sedimentary phosphorus release on a multi-year scale

Astrid Hylén[1*], Sebastiaan J. van de Velde[2,3,4], Mikhail Kononets[1], Mingyue Luo[5], Elin Almroth-Rosell[6], Per O. J. Hall[1]

[1]Department of Marine Sciences, University of Gothenburg, Box 461, 405 30 Gothenburg, Sweden
[2]Department of Earth and Planetary Sciences, University of California, Riverside, CA 92521, USA
[3]Bgeosys, Geoscience, Environment & Society, Université Libre de Bruxelles, 1050 Brussels, Belgium
[4]Operational Directorate Natural Environment, Royal Belgian Institute of Natural Sciences, 1000 Brussels, Belgium
[5]Department of Analytical, Environmental and Geo-Chemistry, Vrije Universiteit Brussel, 1050 Brussel, Belgium
[6]Oceanographic Research, Swedish Meteorological and Hydrological Institute, 426 71 Västra Frölunda, Sweden

*Correspondence to*: Astrid Hylén (astrid.hylen@marine.gu.se)

**Abstract.** Phosphorus fertilisation (eutrophication) is expanding oxygen depletion in coastal systems worldwide. Under low-oxygen bottom-water conditions, phosphorus release from the sediment is elevated which further stimulates primary production. It is commonly assumed that re-oxygenation could break this 'vicious cycle' by increasing the sedimentary phosphorus retention. Recently, a deep-water inflow into the Baltic Sea created a natural in situ experiment that allowed us to investigate if temporary re-oxygenation stimulates sedimentary retention of dissolved inorganic phosphorus (DIP). Surprisingly, during this three-year-long study, we observed a transient but considerable increase, rather than a decrease, in the sediment efflux of DIP and other dissolved biogenic compounds. This suggested that the oxygenated inflow elevated the organic matter degradation in the sediment, likely due to an increase in organic matter supply to the deeper basins, potentially combined with a transient stimulation of the mineralisation efficiency. As a result, the net sedimentary DIP release per m$^2$ was 56-112 % higher over the years following the re-oxygenation than before. In contrast to previous assumptions, our results show that inflows of oxygenated water to anoxic bottom waters can increase the sedimentary phosphorus release.

## 1 Introduction

Eutrophication is one of the main causes of oxygen depletion in coastal systems worldwide (Breitburg et al., 2018; Diaz and Rosenberg, 2008). Excess input of nutrients from land stimulates primary production, resulting in a higher delivery of organic matter to deeper coastal waters and the seafloor (Breitburg et al., 2018; Middelburg and Levin, 2009). Oxygen is consumed as

this organic matter is degraded and in severe cases, anoxia can develop. Depletion of oxygen can lead to decreasing biodiversity and thereby a loss of important ecosystem functions and services (Breitburg et al., 2018; Middelburg and Levin, 2009). One important example is food supply, since many coastal areas are key sites for fishing and aquaculture (Diaz and Rosenberg, 2008).

The Baltic Sea is strongly affected by eutrophication and oxygen depletion. The area has been argued to be a "time machine" for future coastal areas worldwide, due to its long history of anthropogenic disturbances and impacts of climate change (Reusch et al., 2018). As such, understanding the feedback mechanisms between eutrophication and oxygen depletion in the Baltic Sea can help us understand the future of our coastal zones. The central parts of the Baltic Sea have been naturally oxygen-depleted for thousands of years due to a strong water-column stratification, a long water residence time and limited deep-water renewal

(Zillén et al., 2008). However, high inputs of nutrients from land have severely worsened the situation and the areal extent of benthic oxygen depletion has increased 10 times in size over the last 115 years (Carstensen et al., 2014). Despite improvements in wastewater treatment and decreased nutrient input from agriculture, the oxygen-depleted area in the Baltic Sea has not decreased in size (Reusch et al., 2018).

The low-oxygen conditions in the Baltic Sea are largely sustained by enhanced recycling of dissolved inorganic phosphorus

(DIP, mainly phosphate) from sediments with oxygen-depleted overlying water (Vahtera et al., 2007). This 'vicious cycle' has been suggested to drive enhanced recycling of DIP; high sedimentary release of DIP fuels extensive growth of cyanobacteria and other phytoplankton which fuel microbial respiration and oxygen consumption as they are degraded, resulting in increased oxygen depletion and further release of DIP (Vahtera et al., 2007). The elevated release of DIP from oxygen-depleted sediments is mainly a consequence of two factors. Firstly, iron (Fe) and (to some extent) manganese (Mn)

oxides, which adsorb and retain DIP in the sediment under oxic conditions, are reduced and solubilised in anoxic environments (Jilbert and Slomp, 2013; Yao and Millero, 1996). Secondly, P is preferentially regenerated from organic matter relative to nitrogen and carbon, which causes an elevated sedimentary DIP release in environments lacking Fe and Mn oxides (Jilbert et al., 2011; Steenbergh et al., 2011, 2013). In anoxic environments, DIP is thus released back to the water column instead of being retained in the sediment where it eventually could be buried as authigenic P minerals (Slomp et al., 1996). Due to high

sedimentary DIP concentrations there is some formation of authigenic P phases also in the anoxic areas of the Baltic Sea (Jilbert and Slomp, 2013), but the overall burial efficiency of P in this environment is low (0.2 – 12 %, compared to > 14 % in oxic parts of the Baltic Sea; Viktorsson et al., 2013).

It is commonly assumed that re-oxygenation of anoxic sediments would reduce the release of DIP (Ruttenberg, 2003; Stigebrandt, 2018), which could break the vicious cycle. Intrusions of large volumes of salty, oxygenated water from the North

Sea (Major Baltic Inflows; MBIs), can temporarily bring new oxygen to the anoxic areas of the Baltic Sea (Liblik et al., 2018). In the winter of 2014-2015, the largest MBI since 1951 occurred and previously long-term anoxic areas were oxygenated (Liblik et al., 2018). Studies conducted shortly after the inflow did indeed show a lowered DIP release from the sediment, but the total DIP release only decreased with 5-23 % on a basin-wide scale (Hall et al., 2017; Sommer et al., 2017) and sequestration

of P in the surface sediment was low (Hermans et al., 2019b). It has so far been unknown how the sedimentary P dynamics evolved after anoxia was re-established.

Here we present a time series study from the Eastern Gotland Basin (EGB, Fig. 1a) in the Baltic Sea, a long-term anoxic basin that was temporarily oxygenated by the 2014-2015 MBI. To investigate the effect of the oxygenation event on the sedimentary P cycle, we conducted in situ measurements of sediment-water fluxes and collected pore-water and solid-phase samples during three years (2016-2018). By combining our results with data from 2008 (Nilsson et al., 2019; Viktorsson et al., 2013), 2010 (Nilsson et al., 2019; Viktorsson et al., 2013) and 2015 (Hall et al., 2017), we construct a unique time series from prior to until after the inflow and show that contrary to previous assumptions, the inflow of oxygenated water to this anoxic area increased the sedimentary P release on a multi-year scale.

## 2 Materials and Methods

### 2.1 Site characteristics

The EGB is the largest sub-basin of the Baltic Sea, situated between Latvia and the Swedish island Gotland (Fig. 1a). The water column is normally hypoxic below the halocline at 60-80 m depth and anoxic at the seafloor (maximum depth 249 m). The MBI that occurred in the winter of 2014-2015 reached the EGB in March 2015, resulting in a more oxygenated but still hypoxic (40-60 µM) water layer below 130-140 m depth (Fig. 1b,c; Hall et al., 2017; Liblik et al., 2018). The oxygen had been consumed by the end of 2015, but a new inflow oxygenated the area again between January and May 2016 (Liblik et al., 2018). In 2017, several smaller inflows also brought oxygen to the 80-170 m depth layer (Liblik et al., 2018).

During yearly expeditions to the EGB in April 2016-2018, we visited three stations along a depth transect (Fig. 2a, Table A1). The stations are situated below the halocline and were oxygenated by inflows at different points during the sampling period. The sediment is rich in organic matter (organic carbon 10-15 % dry weight) and has a high surface porosity (0.95-0.98; Nilsson et al., 2019; van de Velde et al., 2020b). Station D (130 m) is situated at the top boundary of the oxygenated layer. It was anoxic during the sampling in 2016 (Fig. 2c), but monitoring data from the Swedish Meteorological and Hydrological Institute (SMHI, Fig. A1; SMHI, 2021) and previous studies (Dellwig et al., 2018; Hermans et al., 2019b; Sommer et al., 2017) from the eastern part of the EGB suggest that this station might have been temporarily oxygenated by the MBI prior to our visit. Station D received some oxygen with the smaller inflow in 2017, but was anoxic again in 2018. Station E (170 m) and F (210 m) were oxygenated by the MBI (Fig. 2c). The oxygen at station F had been consumed already in 2017, while it decreased stepwise at station E. Hydrogen sulphide was only detected in the bottom water at station D in 2016 (Broman et al., 2020; Marzocchi et al., 2018).

For clarity, the remainder of this paper will focus on the results from station D and F. These stations represent the least and most affected sites, respectively. The results from station E, which are consistent with the results from station F, are presented in Figs. A2-A4.

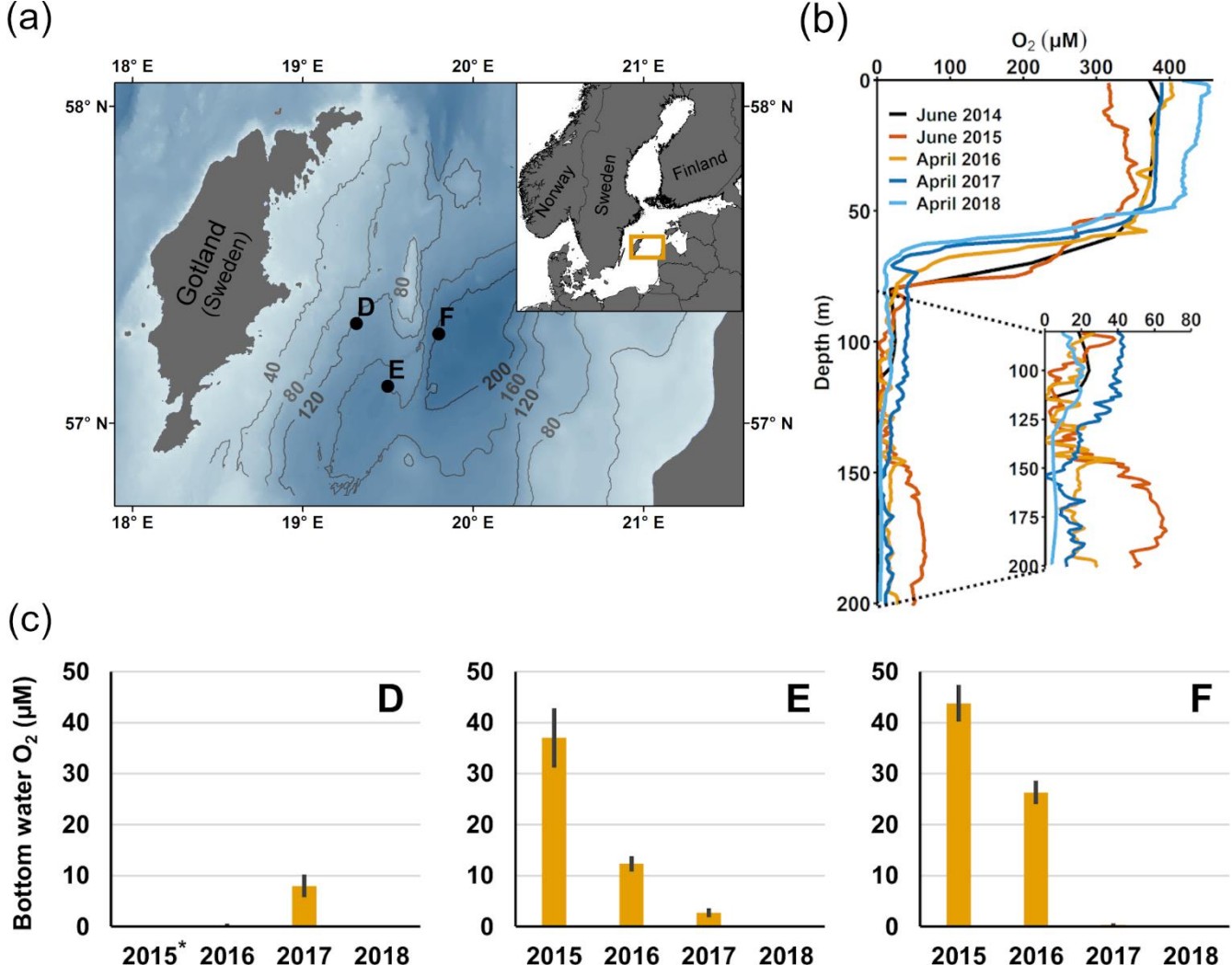

**Figure 1.** Description of sampling sites and oxygen conditions. (a) Map of the sampling sites in the Eastern Gotland Basin, Baltic Sea. Bathymetric data from the Baltic Sea Bathymetry Database (v. 0.9.3), the map was created using the software ArcMap$^{TM}$ (v. 10.6) by Esri. (b) Water-column profiles of oxygen ($O_2$). Data from SMHI (SMHI, 2021; 2014, station BY15), Hall et al. (2017; 2015, station F) and van de Velde et al. (2020b; 2016-2018, station F). (c) Bottom-water oxygen ($O_2$) concentrations at stations D-F measured by optodes on the benthic chamber lander (average ± standard deviation) ~0.2 m above the sediment. The detection limit was 0.5 µM. Data from Hall et al. (2017) and van de Velde et al. (2020b). *no measurement.

## 2.2 Water-column and benthic in situ flux measurements

Upon arrival at the stations, a CTD (SBE 911, Sea-Bird Scientific) with a high-accuracy oxygen sensor (SBE 43, Sea-Bird Scientific) was deployed to record water-column profiles of salinity, temperature and oxygen (van de Velde et al., 2020b).

Sediment-water fluxes of nutrients and dissolved inorganic carbon (DIC) were subsequently measured using the Gothenburg benthic chamber lander (Kononets et al., 2021). Deployment of the lander during these samplings has been described in detail

elsewhere (Hall et al., 2017; van de Velde et al., 2020b). Briefly, the lander carried four incubation chambers incubating 400 $cm^2$ of sediment with overlying water each. At pre-set times, nine syringes per chamber sampled the overlying water. Sensors monitored conductivity, temperature and oxygen inside and outside the chambers. Chambers that initially contained oxygen did not become anoxic over the course of the incubations. Chamber volumes were calculated from the decrease in salinity following the injection of a small, known volume of Milli-Q water. Sediment and water were incubated for 14 hours.

Immediately after lander recovery, the syringes were emptied and samples were filtered through pre-rinsed 0.45 µm cellulose acetate filters (Sartorius). The lander was deployed twice per station except for two occasions (F 2017, E 2018) when time constraints only allowed one deployment.

Calculation of benthic nutrient and DIC fluxes are described in van de Velde et al. (2020b). The fluxes from 2008, 2010 and 2015 (Hall et al., 2017; Nilsson et al., 2019; Viktorsson et al., 2013) were measured using the same lander systems as in this

study and have been re-evaluated to assure that the same evaluation method is used and thus the results are comparable. In short, a linear regression model, simple or quadratic, was fitted to data of the concentration change over time in the chamber water. Models were selected based on their Akaike information criterion score, corrected for small sample sizes. After verifying model and data quality, the flux was calculated by multiplying the chamber water height with the initial slope of the regression line. About 5 % of the regression models gave slopes for which $p > 0.05$. After visual inspection of the data, all fluxes were

kept in the data set in order to not overestimate the average fluxes by excluding low flux values.

The effects of year and station (fixed factors) on sediment-water fluxes were tested with a type III ANOVA, with lander deployment as a nested factor within year and station. Before analysis, flux data were inspected visually for homogeneity of variance and were $log_{10}$ transformed. Significant effects of year and station were investigated using Student-Newman-Keuls (SNK) tests. All tests were two-sided with a significance level of 0.05. Analyses were conducted in the statistical software R

(R Core Team, 2020), using the functions 'lm' and 'anova' from the standard CRAN:stats package and 'SNK.test' from the CRAN:agrocolae package (de Mendibur, 2020).

Deployment of benthic chamber landers is a commonly used technique to measure in situ fluxes and avoid potential manipulation artefacts during ex situ incubations. However, some caution should be taken when interpreting the absolute values of the measured fluxes. For example, during incubation of sediment from sites with stagnant water, erosion of the

diffusive boundary layer by stirring of bottom water can increase the sedimentary oxygen uptake by up to 30 % (Glud et al., 2007; Hall et al., 1989). It is likely that changes to the thickness of the benthic boundary layer also affect benthic releases of DIC and nutrients, but this is yet to be confirmed. Nevertheless, the fluxes in this study may be slightly overestimated. However, during deployment the stirring mechanism was made to resemble the natural bottom currents as much as possible and we expect the impact on our measured fluxes to be smaller than 30%. Furthermore, such an impact on the fluxes would

only affect absolute values, not patterns between years.

## 2.3 Sediment sampling

Pore water was extracted from sediment cores (9.9 cm inner diameter) collected with a multiple corer. At each station, one core was sectioned in a portable glove bag ($N_2$ atmosphere; Aldrich AtmosBag, Sigma-Aldrich). The cores were sliced at 0.5 cm resolution from 0 to 2 cm depth, at 1 cm resolution between 2 and 6 cm depth, and in 2 cm slices from 6 to 20 cm depth. The sediment was transferred into centrifuge tubes into which Rhizon samplers (Rhizosphere research products) were inserted for collection of pore water.

Samples for solid-phase speciation of P were collected at station D and F in 2016 and 2017. Due to logistical constraints, samples were not collected on the other occasions. Sediment was collected using a modified box corer (Blomqvist et al., 2015) from which 6 cores were taken using transparent PVC core liners (6 cm inner diameter; 30 cm long). Two cores per station were sectioned in a portable glove bag ($N_2$ atmosphere; Captair Field pyramid, Erlab) under constant oxygen monitoring (Portable oxygen analyser 3110 with trace oxygen sensor, Teledyne). The cores were sliced at 0.5 cm resolution from 0 to 3 cm depth, at 1 cm resolution between 3 and 10 cm depth, and in 2 cm slices from 10 to 14 cm depth. The sediment was transferred into polypropylene centrifuge tubes (TPP), centrifuged at 2500g for 10 min (Sigma 3-16L, Sigma) and was returned to the anoxic glove bag. After removal of the overlying pore water, the solid phase was freeze-dried and stored in an aluminium bag under $N_2$ atmosphere for later speciation of P. In 2016, the pore water was filtered (0.42 µm cellulose acetate filters, Chromafil Xtra) and acidified with 100 µL mL$^{-1}$ $H_2SO_4$ (1 M) for later analysis as a control of our pore-water sampling procedure.

## 2.3 Bottom-water and pore-water analysis

Nutrient samples from the lander and multiple corer pore-water samples were determined by segmented flow colorimetric analysis (Alpkem Flow Solution IV, OI Analytical; Koreleff, 1983). The analytical precisions were 3 % for DIP, 3 % for dissolved inorganic nitrogen (DIN) and 2.5 % for dissolved silica (DSi). In 2016, pore-water DIP samples from the box corer were determined by segmented flow colorimetric analysis (QuAAtro, SEAL Analytical; Aminot et al., 2009). The precision was < 4 %.

DIC concentrations in the lander samples were determined on-board the ship by non-dispersive infrared spectrometry (LI-COR 6262) after acidification with phosphoric acid (Nilsson et al., 2019). Repeated measurements of certified reference material (CRM from Dickson Laboratory, Scripps Inst. of Oceanography) were conducted to obtain a two-point calibration and correction for potential drift in the system. The analytical precision was 0.2 % (relative standard deviation).

## 2.4 Sequential phosphorus extractions

A subsample of ~100 mg sediment from each depth was used for sequential P extraction (SEDEX; Ruttenberg, 1992; Slomp et al., 1996). The SEDEX procedure separates total sedimentary P into five operationally defined fractions; exchangeable or loosely sorbed P ($P_{ex}$), easily reduced or reactive Fe bound P ($P_{Fe}$), authigenic P ($P_{auth}$), detrital P ($P_{det}$) and P associated with organic matter ($P_{org}$). All extractions were performed at room temperature and under constantly agitated conditions; extraction solutions and times are shown in Table 1. After each extraction, the sample was centrifuged (2500 g for 10 min) and the supernatant was filtered (0.45 µm cellulose acetate) and subsequently analysed spectrophotometrically. The first two extraction steps were performed under oxygen-free conditions. The $P_{Fe}$ fraction was analysed by inductively coupled plasma − mass spectrometry (ICP-MS), since the reagent interferes with the spectrophotometric colour reagent. Performance of the method was monitored by analysing one sample in triplicate during each batch of extractions (to assess the repeatability by calculating the relative standard deviation), and by analysing subsamples of one sample with every batch of extractions (to assess reproducibility by calculating the relative standard deviation). For each extraction step, repeatability was around 5% and reproducibility was around 10 %. Calculations of sedimentary inventories of different P fractions are described in appendix B.

**Table 1.** Specifics for the sequential extractions of phosphorus. Brief summary of the reagents, methods and times used for SEDEX. Each step was followed by a wash with 1M $MgCl_2$ (pH 8) for 30 minutes, to avoid re-adsorption of DIP.

| Fraction | Extraction solution | Time |
|---|---|---|
| $P_{ex}$ | 1 M $MgCl_2$ | 30 min |
| $P_{Fe}$ | 12 g sodium dithionite in 480 mL sodium acetate (1 M) + 60 mL sodium bicarbonate (1 M) | 8 h |
| $P_{auth}$ | 300 mL sodium acetate (1 M) + 1700 mL acetic acid (1 M) | 6 h |
| $P_{det}$ | 1 M HCl | 24 h |
| $P_{org}$ | Combustion at 550°C | 2 h |
| | 1 M HCl | 24 h |

 ## 3 Results and Discussion

### 3.1 Enhanced sedimentary P release after the inflow

Sediment-water fluxes of DIP measured in situ showed significant changes with time following the MBI (Fig. 2a, Table A2, Table A3). At station F, the DIP efflux decreased markedly in 2015 (by 81 %) compared to the 2008 and 2010 (pre-inflow) values, and even turned into an uptake in about half of the chambers (Hall et al., 2017). In 2016, however, the DIP flux was

about 4.5 times higher than before the inflow, after which it decreased stepwise in 2017 and 2018, returning to pre-inflow values. Although the DIP fluxes were highest at station F, similar patterns were observed at all stations (Fig. A2, Table A2, Table A3). Intriguingly, the DIN fluxes followed the same pattern as the DIP fluxes, and similar trends were seen for DIC and DSi.

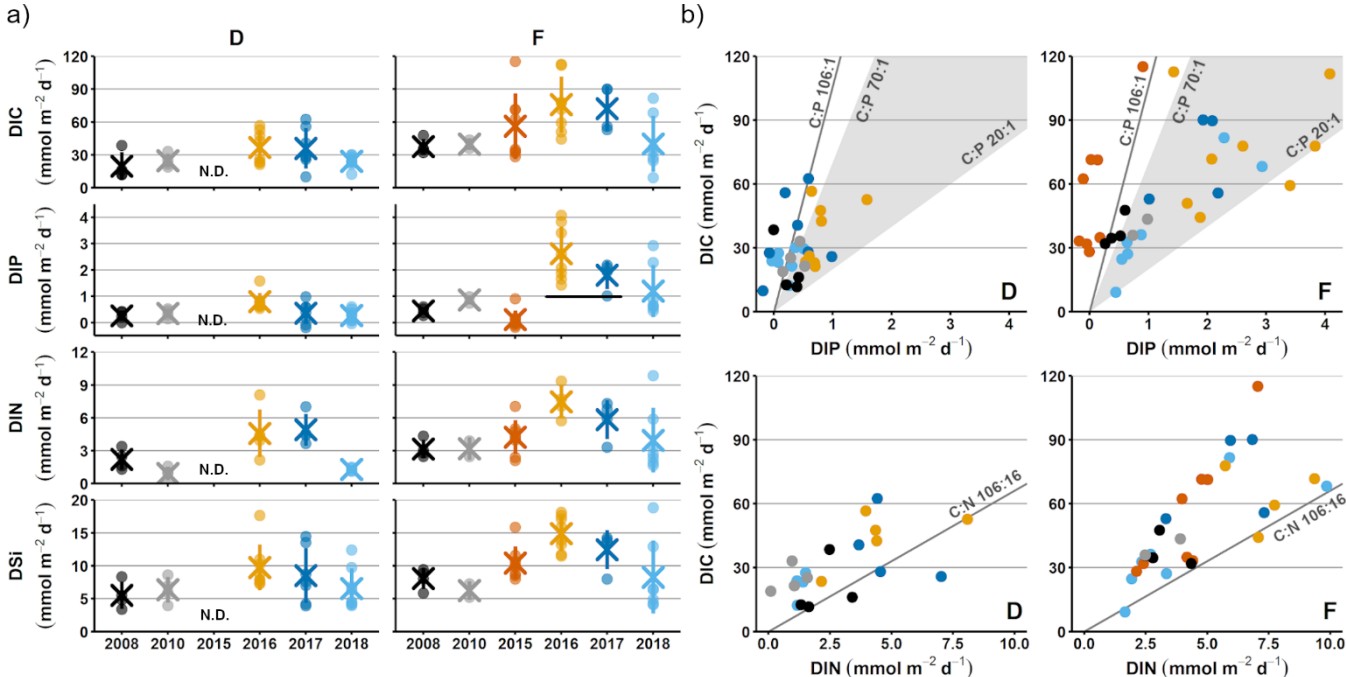


**Figure 2.** Sediment-water fluxes measured with the benthic chamber lander. Flux data are from before the inflow (2008, 2010; Hall et al., 2017; Nilsson et al., 2019; Viktorsson et al., 2013), right after the inflow (2015; Hall et al., 2017) and post inflow (2016-2018; this study). (a) Changes in fluxes with time. Small circles mark individual chamber measurements, crosses mark station averages and lines represent station standard deviations. The black line in the DIP-graph for station F in 2016-2017 marks the expected flux for those years, based on the

decrease in solid phase inventory. N.D. = no data. (b) Fluxes of DIC versus DIP and DIN. The grey line represents the Redfield ratios of 106:1 for DIC:DIP and 106:16 for DIC:DIN, the shaded area marks the DIC:DIP ratios between 20:1 and 70:1. Colour coding the same as in a).

The decrease in DIP flux at station F in 2015 temporarily raised the DIC:DIP flux ratio above the Redfield ratio (Redfield,

1958) of 106:1 (Fig. 2b; Hall et al., 2017). As the DIC:DIN ratio did not change, the lowered DIP flux was most likely due to

trapping of P by Fe and Mn oxides, formed at the sediment surface as a result of the oxygenation. When anoxia returned in 2016, $P_{Fe}$ that had been temporarily trapped during oxic conditions was expected to be released and increase the sedimentary DIP efflux, which would lower the DIC:DIP flux ratio compared to pre-MBI conditions. However, the DIC:DIP ratio only decreased to 30-40 in 2016-2018, which is within the range of what is normally observed in the anoxic parts of the EGB (~20-
70; Sommer et al., 2017; Viktorsson et al., 2013; Fig. 2b). These low DIC:DIP flux ratios are the result of preferential regeneration of P from organic matter, causing an elevated sedimentary DIP release in the absence of Fe oxides (Jilbert et al., 2011; Steenbergh et al., 2011, 2013), and release of DIP adsorbed to Fe oxides that are shuttled from shallower areas (Dellwig et al., 2010). The consistent DIC:DIP ratio in all years apart from 2015 (right after the MBI) implies that other factors than release of temporarily trapped $P_{Fe}$ contributed to the increased benthic DIP efflux at station F in 2016-2017. An intriguing clue
to what could have caused the elevated DIP fluxes was given by the fluxes of other biogenic solutes , which followed the same pattern of temporarily increased fluxes (Fig 2a, Table A2, Table A3). The co-variation suggests that a common factor affected all solute fluxes, hinting towards an increase in the organic matter mineralisation rate and/or organic matter supply. The pattern in fluxes over time observed at station F was also significant at station D, but the fluxes were lower and the pattern was less pronounced.. A sedimentary uptake of DIP was measured in two out of seven chambers in 2017, suggesting that the
oxygenation of the sediment was weak and heterogeneous. Significant differences in fluxes between lander deployments indeed suggest that the sediment environment in the entire area was heterogeneous (Table A2). Station D is situated at the edge of the area that was affected by the MBI and the oxygenation there was less strong (Fig. 1c). The conditions at station D can thereby be assumed to represent a lower boundary for effects from the inflow. At both stations D and F, the DIC:DIN flux ratios were around 10-12 (Fig. 2b) rather than the Redfield ratio of 6.6 (106:16). The flux ratios in this study agree well with
carbon:nitrogen ratios at the sediment surface and in sinking particles in the EGB (Cisternas-Novoa et al., 2019; Nilsson et al., 2021).

## 3.2 Sedimentary phosphorus dynamics after the inflow

Sediment profiles from all stations showed clear signs of oxygenation after inflows (Fig. 3a, Fig. 4a). At station F, the pore-
water DIP displayed distinct subsurface peaks in 2016 (Fig. 3a). This type of profile indicates production of DIP in the upper sediment layers (Ruttenberg, 2003), probably caused by a sudden influx of P in the form of reactive organic matter or P adsorbed to Fe and Mn oxides. In contrast, pore-water DIP gradually accumulated with depth in 2017 and 2018. Both the $P_{ex}$ and $P_{Fe}$ concentrations were higher in 2016 than in 2017 (Fig. 3a; other P fractions in Fig. A4). This suggests that there was a transient enrichment of these fractions in 2016 that had disappeared in 2017, which would explain the DIP peak.
The sequestration of P after oxygenation was calculated as the excessive inventories (Fig. B1) of $P_{ex}$ and $P_{Fe}$, i.e. the integrated enrichment of these fractions compared to their background concentrations. The excess inventory of $P_{ex}$ and $P_{Fe}$ suggest that the net P retention at station F by the sampling occasion in 2016 was 4.6 – 6.4 mmol P m$^{-2}$ (Fig. 3b), consistent with a study from the eastern part of the EGB where 12 mmol P m$^{-2}$ was estimated to have been retained due to the MBI (Hermans et al.,

2019b). By the sampling occasion in 2017, the net excess inventory had decreased to 3.1 – 3.5 mmol P m$^{-2}$. This means that
1.1 – 3.3 mmol P m$^{-2}$ was released from temporary storage in the sediment between the 2016 and 2017 samplings. As no
increase was seen in the $P_{auth}$ fraction (Fig. A4), it appears that most of this P was lost to the overlying water. A release of 3.3
mmol P m$^{-2}$ in one year would lead to an average sedimentary efflux of 0.01 mmol m$^{-2}$ d$^{-1}$. However, the benthic DIP flux
measured in 2016-2017 was 1.2 – 2 mmol m$^{-2}$ d$^{-1}$ higher than before the inflow (Fig. 2a). As suggested by the DIC:DIP ratio,
another source of P must have contributed to the DIP flux at station F in 2016-2018.

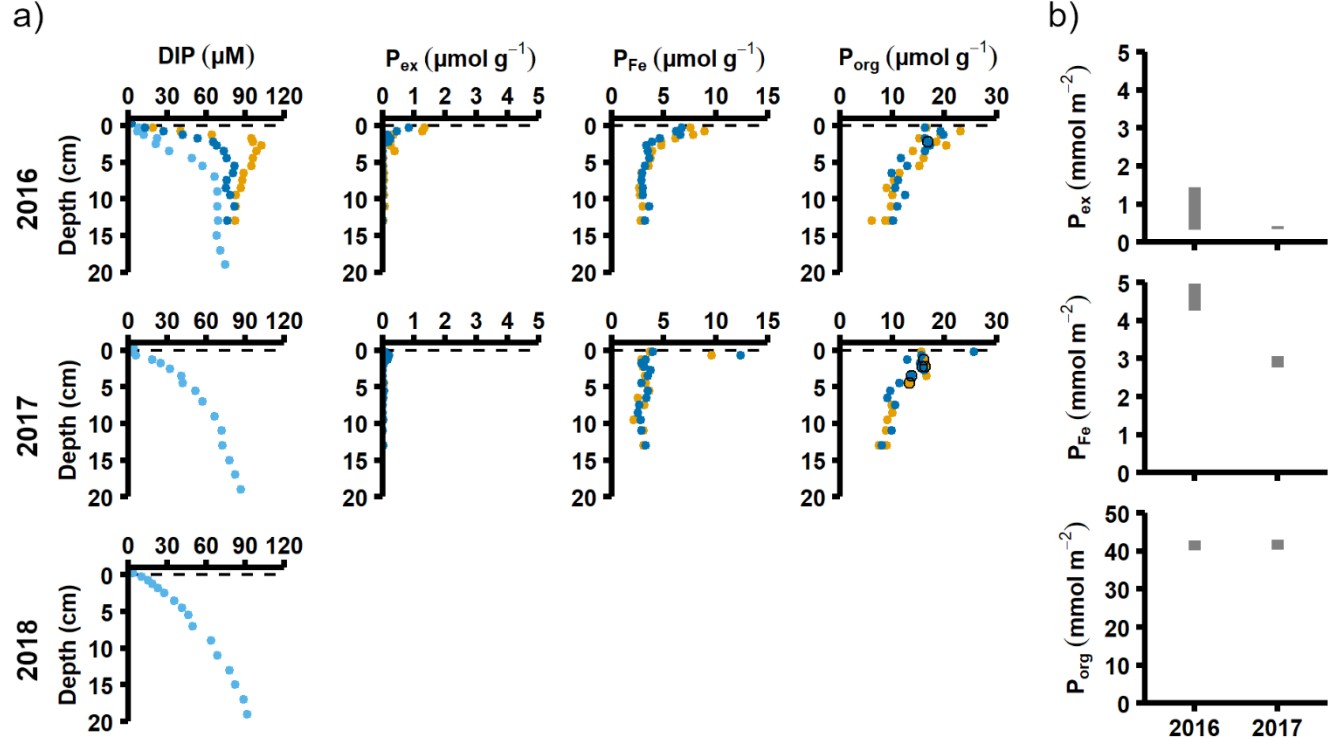

**Figure 3.** Pore-water and sediment solid phase data from station F. (a) Profiles of pore-water DIP, solid-phase exchangeable or loosely-sorbed P ($P_{ex}$), easily-reduced or reactive iron-bound P ($P_{Fe}$) and organic P ($P_{org}$). Profiles with light blue markers are from big sediment cores (inner diameter 9.9 cm); profiles with dark blue and orange markers are from replicate small sediment cores (inner diameter 6 cm). Markers with black edges are missing data that have been replaced with average concentrations from the slices directly above and below (see appendix B). (b) Excess inventories (min – max range) of $P_{ex}$ and $P_{Fe}$, and inventories of $P_{org}$ in the top 5 cm.

Signs of oxygenation were also seen at station D. In 2016 and 2018, the pore-water DIP accumulated with sediment depth.
However, the profile was very different in 2017 and displayed a peak at 4 cm depth, suggesting an increased release of reactive
P (Ruttenberg, 2003). Subsurface peaks of $P_{ex}$ and $P_{Fe}$ at station D were visible in one of the cores from 2016. The variation
between replicate cores in 2016 raises the question whether changes in pore-water DIP, $P_{ex}$ and $P_{Fe}$ at station D were caused
by spatial variability, or if there was a change over time in response to oxygenation. However, the changes in sedimentary P

profiles and fractionation follow what would be expected if station D had been partially affected by the MBI; a likely scenario based on oxygen monitoring data (Fig. A1; SMHI, 2021). The 2017 inflow had a smaller effect on the sediment geochemistry

at station D, since no peaks were present in the cores collected that year. At station D, the $P_{ex}$ + $P_{Fe}$ excess inventory change between the 2016 and 2017 samplings ranged from +1.3 to -21.5 mmol m$^{-2}$ (Fig. 4b). As there was no increase in the authigenic P fraction (Fig. A4), it is likely that this P was released to the water column as DIP. An excess P release at station D of 21.5 mmol m$^{-2}$ would only have increased the DIP flux by 0.06 mmol m$^{-2}$ d$^{-1}$ during 2016-2017, however. The DIP flux at station D did indeed not change substantially over the sampling (Fig 2).


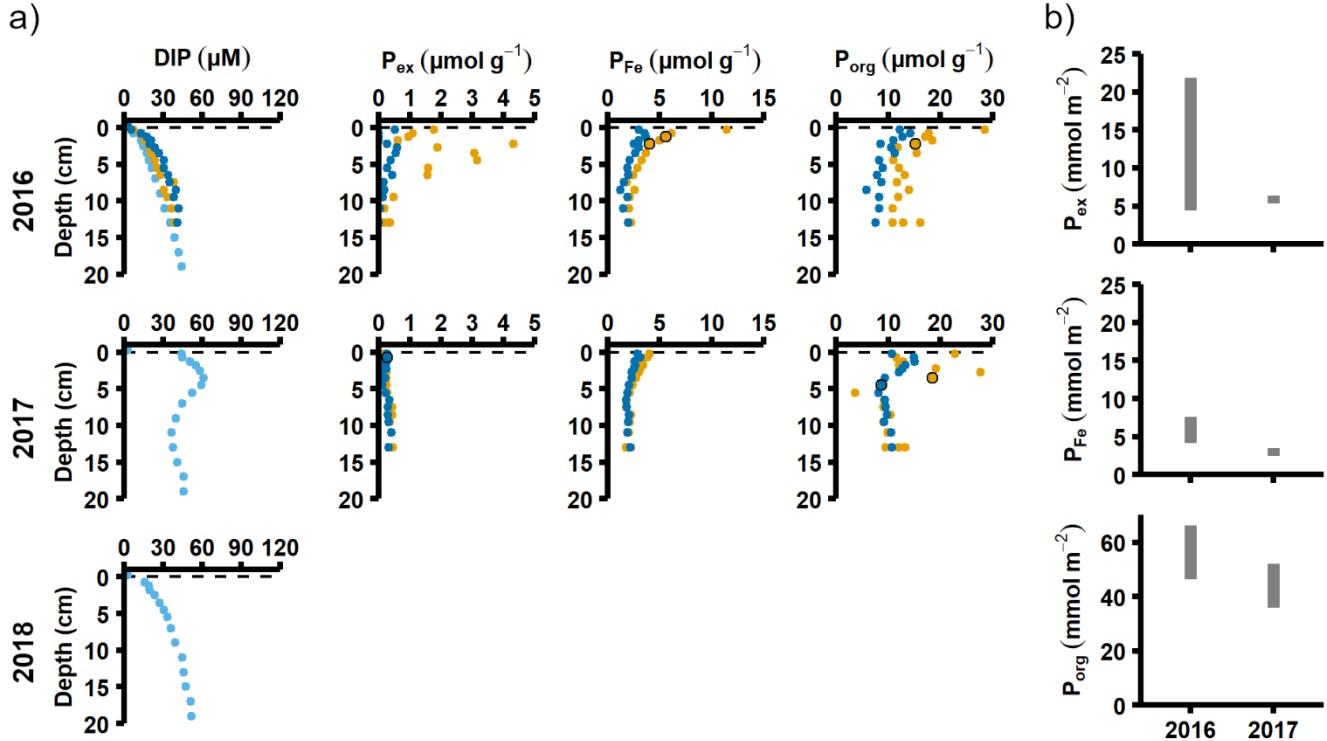

**Figure 4.** Pore-water and sediment solid phase data from station D. (a) Profiles of pore-water DIP, solid-phase exchangeable or loosely-sorbed P ($P_{ex}$), easily-reduced or reactive iron-bound P ($P_{Fe}$) and organic P ($P_{org}$). Profiles with light blue markers are from big sediment cores (inner diameter 9.9 cm); profiles with dark blue and orange markers are from replicate small sediment cores (inner diameter 6 cm).

Markers with black edges are missing data that have been replaced with average concentrations from the slices directly above and below (see appendix B). (b) Excess inventories (min – max range) of $P_{ex}$ and $P_{Fe}$, and inventories of $P_{org}$ in the top 5 cm.

The in situ measured DIP fluxes were considerably higher than diffusive fluxes calculated from pore-water profiles (Table C1). Moreover, the $P_{org}$ inventories in the top 5 cm of the sediment (Fig. 3b, Fig. 4b) could only support the DIP fluxes

measured in 2016-2017 for 59-159 days at station D and 15-24 days at station F. This discrepancy between measured fluxes and calculated diffusive fluxes and sedimentary $P_{org}$ inventories may partly be explained by potential overestimation of

measured fluxes (see section 2.2). However, even in the unlikely case that the measured fluxes overestimated the in situ fluxes by 50 %, the $P_{org}$ inventories would only support the DIP fluxes for 118-318 days at station D and 30-48 days at station F. Alternatively, it is well known that chemical gradients at the sediment-water interface may not be accurately captured by the limited vertical resolution of pore water profiles, resulting in underestimates when calculating diffusive fluxes (Nilsson et al., 2019; Noffke et al., 2012; Sundby et al., 1986; van de Velde et al., 2020b). Since the sediment at stations D and F had such high porosities (~0.98; van de Velde et al., 2020b; Fig A5), it is also possible that the topmost fluffy and organic rich layer was not captured during sampling and processing of sediment cores. Solute fluxes caused by processes in this sediment layer would have been detected in the benthic chamber lander incubations, however. The high in situ measured fluxes relative to calculated diffusive fluxes and sediment $P_{org}$ inventories thus suggests that processes occurring at the sediment-water interface were the main driver behind the benthic fluxes, whereas sedimentary inventories played a minor role. Previous studies have indeed suggested that organic matter mineralisation is rapid and efficient in the deep basins of the EGB (Nilsson, 2018; Nilsson et al., 2021), indicating that it mostly takes place at the sediment surface. In addition to vertical deposition of phytoplankton material, there is an extensive horizontal shuttling of material from shallow to deeper parts of the EGB (Christiansen et al., 2002; Jonsson et al., 1990; Leipe et al., 2011) which is thought to support about half of the benthic organic matter mineralisation in the deep basin on an annual basis (Nilsson et al., 2021).

If the topmost sediment layer was not accurately captured during sampling, the calculated $P_{ex}$ and $P_{Fe}$ inventories are likely also too low. This does not necessarily mean that the change in inventories between years is underestimated, if equally large fractions of $P_{ex}$ and $P_{Fe}$ were missed both years. Much of the change in $P_{ex}$ and $P_{Fe}$ between 2016 and 2017 is expected to have taken place at the sediment-water interface, however, suggesting that the inventory change was in fact larger than estimated here. Nevertheless, the constant DIC:DIP flux ratio in all years but 2015, together with similar flux patterns for multiple biogenic solutes, suggest that the contribution from $P_{ex}$ and $P_{Fe}$ to the total DIP flux was, in fact, small.

### 3.3 Increased mineralisation after the inflow

Our data suggest that the elevated fluxes of DIP and other dissolved biogenic elements were caused by increased rates of sedimentary organic matter mineralisation. Changes in sedimentary mineralisation rates can be caused by an enhanced input of reactive organic matter to the sediment (for example after a spring bloom or by enhanced organic matter export out of the water column) or by an increase of the intrinsic mineralisation rate of the organic matter already present in the sediment. We will discuss these different possibilities below.

### 3.3.1 Seasonality, primary production and benthic fauna

Data for this study were collected during different seasons (2008, 2010: August/September; 2015: early July; 2016-2018: April), so some variation in fluxes due to seasonality is expected. However, even though DIC and nutrient effluxes should have been highest in 2008, 2010 and 2015 due to sedimentation of fresh phytoplankton material (Berelson et al., 2003), similar

flux values were measured in 2018. Instead, the highest fluxes were recorded during the spring samplings (2016-2018, Fig.

2a, Table A3). The differences in timing of the sampling relative to the spring bloom could still be a potential cause. However, both the lack of clear surficial peaks in the chlorophyll a sediment profiles (Fig. D1) and monthly water-column monitoring data of phytoplankton and nutrient concentrations from monitoring data (Fig. A5; SMHI, 2021) suggest that all samplings in 2016-2018 were conducted before the spring bloom. In fact, the year when the sampling was most likely to coincide with the spring bloom according to the SMHI data was in 2018, which was the year with the lowest fluxes of all spring samplings. It

thus seems highly unlikely that seasonality was driving the observed flux pattern.

The MBI could also have pushed up deep water to the surface, thereby providing nutrients to phytoplankton blooms. No increase in primary production was observed, however, during either the spring bloom or summer bloom in the years after the MBI (Johansen and Skjevik, 2016-2017). Instead, the variation of fluxes closely follows the pattern of oxygenation, with higher fluxes just after the oxygenation and a gradual decrease to pre-MBI values once all oxygen had been consumed in the

bottom waters (Fig. 1c, Fig. 2).

Alternatively, organic matter mineralisation rates is known to increase with temperature (Arnosti et al., 1998). The water temperature below the halocline in the EGB does not follow seasonal patterns (Fig. A6; SMHI, 2021); different sampling months (April vs August/September) should thus not have affected the benthic fluxes. The MBI increased the bottom water temperature by ~1°C, however (Fig. A6; Liblik et al., 2018; SMHI, 2021). Despite lower temperatures in 2008 and 2010

compared to 2018, the benthic fluxes were similar during these years (Fig. 2a). Temperature differences are thus unlikely to alone have driven the benthic flux pattern.

Oxygenation can allow re-colonisation of previously anoxic sediments by benthic animals. Their presence would result in sediment reworking (bio-mixing) and bio-irrigation, which are believed to stimulate organic matter degradation (Ekeroth et al., 2016; van de Velde et al., 2020a). Two individuals of *Bylgides sarsi* were found near our sampling sites in 2015 (Hall et

al., 2017; Stigebrandt et al., 2018). However, all sediment retrieved from the newly oxygenated area during our sampling was laminated and there were no signs of colonisation by animals, consistent with other studies conducted in the EGB in 2016 (Hermans et al., 2019a). It is thus likely that the two specimens of *Bylgides sarsi* had arrived with the inflowing water (Stigebrandt et al., 2018). The anoxic water layer at 100-140 m depth would further have created a barrier that prevented recolonization of the newly oxygenated sediments. We therefore exclude animal activity as an explanation for the enhanced

sediment-water fluxes after the MBI.

### 3.3.2 Enhanced organic matter input

The oxygenation of the bottom waters following a MBI could potentially lead to enhanced reactive particulate organic matter input to the sediment. In oxygen minimum zones, particle fluxes are normally higher than in surrounding oxic waters, possibly

because of lower particle mineralisation rates as a result of less energetic electron acceptors or loss of zooplankton that consume particles (Le Moigne et al., 2017). However, a study conducted in the EGB in the summer of 2015 suggested that the particle

sedimentation rate and freshness of the organic material reaching the sediment increased due to the MBI (Cisternas-Novoa et al., 2019). In the newly oxygenated EGB, the flux of particulate organic carbon (POC) increased by 18 % between 40 m and 180 m depth, while it decreased by 28 % at a permanently anoxic site nearby (Cisternas-Novoa et al., 2019). The cause of the increased POC transport in the EGB was thought to be the formation of Mn oxides in the oxygenated water column, which scavenged organic matter while sinking (Cisternas-Novoa et al., 2019). Several other studies have confirmed that the MBI led to a considerable formation and sedimentation of Mn oxides in the water column of the EGB (Dellwig et al., 2018; Hermans et al., 2019b; van de Velde et al., 2020b). Enhanced transport of organic matter to the sediment via Mn oxides thus have led to a higher deposition and increased benthic organic matter degradation rates. According to the numbers presented by Cisternas-Novoa et al. (2019), however, the increased vertical POC flux after oxygenation would still have remained relatively low.

In contrast, lateral transport could have been a substantial source of POC and associated elements after the inflow. Measurements in the south-eastern part of the EGB in March 2015 showed increased turbidity close to the seafloor below 140 m depth, as a gravity current generated by the inflow moved along the slope (Schmale et al., 2016). As the inflow moved north, it could thus have enhanced the constantly ongoing lateral transport of relatively fresh organic matter from shallower to deeper areas (Nilsson et al., 2021). Furthermore, the speed of the gravity current generated by the inflow was higher than current speeds generally observed in the area during stagnation periods (Hagen and Feistel, 2004; Schmale et al., 2016). The higher current speed could have resulted in a more intense resuspension and of larger particles (Danielsson et al., 2007), including material that would otherwise have been buried. This mixing of old and new organic matter could additionally have induce a 'priming' effect, where older material is broken down more efficiently in the presence of the fresher material (Bianchi, 2011; van Nugteren et al., 2009).

### 3.3.3 The effect of oxygen

Oxygen can directly contribute to an enhancement of the organic matter mineralisation rate. Increased mineralisation rates under oxic conditions, compared to anoxic conditions, have been suggested for refractory (or aged) and some types of fresh organic matter (Arndt et al., 2013; Bianchi et al., 2018; Hulthe et al., 1998). The reason for this is debated, but factors such as production of different enzymes and higher energy yield during degradation of organic matter under oxic conditions have been proposed (LaRowe et al., 2020). Although oxygen can affect the mineralisation of organic matter, the connection between oxygen and increased DSi fluxes is less clear. It is possible that oxygen promoted the degradation of the organic coating covering diatom frustules, which in turn could enhance the dissolution of biogenic silica (Bidle and Azam, 1999). The elevated DSi fluxes observed here could also indicate that an increased input of organic matter (with which biogenic silica is associated, such as diatom material), rather than the presence of oxygen, was causing the change in fluxes.

Interestingly, a study from the EGB conducted shortly after the MBI found that while the intrusion of oxygen in itself did not enhance the heterotrophic bacterial production in the water column, the removal of hydrogen sulphide did (Piontek et al.,

2019). The same mechanism could be relevant for the surface layer of the sediment, since the MBI also resulted in the removal of hydrogen sulphide right below the sediment-water interface (Marzocchi et al., 2018).

### 3.4 Environmental implications

Following the MBI, there was a substantial elevation of the sedimentary DIP release in the EGB despite oxygenation of the
sediment. Although Fe and Mn oxides were present at the sediment surface (Fig. 3a, Fig. 4a; Dellwig et al., 2018; Hermans et al., 2019b), they did not prevent DIP from escaping to the water column. The oxygen penetration was restricted to the top 2 mm of the sediment, and fluxes of hydrogen sulphide and other reduced compounds from deeper sediment layers appear to have prevented extensive oxidation of the sediment (Hermans et al., 2019b; Marzocchi et al., 2018). In the eastern part of the EGB, the ratio between Fe oxides and $P_{ex} + P_{Fe}$ and was ~1 (Hermans et al., 2019b), suggesting that the adsorption sites on the
Fe oxides were saturated since the lowest ratio Fe oxide:P ratio that has been observed in marine environments is ~2 (Gunnars et al., 2002). Other minerals, such as Mn oxides and carbonates, could also act as sorption sites for phosphate (Kraal et al., 2017; Yao and Millero, 1996). After the MBI, large quantities of Mn oxides were deposited on the sediment (Dellwig et al., 2018; Hermans et al., 2019b; van de Velde et al., 2020b). Hermans et al. (2019b) found clear associations between Mn and P enrichments in the sediment, and stipulated that Mn minerals had an important function in the retention of phosphate, which
could explain the relatively large amount of $P_{ex} + P_{Fe}$ relative to Fe oxides. Nevertheless, the fact that DIP escaped from the sediment to the water suggests either that all adsorption sites were saturated, or that the adsorption process was not efficient (e.g. due to kinetic limitations) relative to the diffusion of DIP toward the water column. Overall, it appears that the sediment of our study site had limited capacity to trap DIP during this short oxygenation event.

To estimate the potential impact on the water column P inventory in the EGB, we propose three scenarios for the total, time-
integrated DIP release from the sediment between March 1 2015 and April 16 2018 (3.13 years, Fig. 5a). The Background scenario describes a situation without the MBI, assuming that the flux equals the average of the fluxes in 2008, 2010 and 2018. The MBI min scenario describes a situation where the flux decreased to the value measured in 2015 as soon as oxygen reached the EGB, stayed at that level until oxygen disappeared and then changed linearly between the measured values. The MBI max scenario describes a situation where the flux begins at the Background value and then changes linearly between the measured
values. At station F, the Background scenario gives a total DIP release of 0.91 mol m$^{-2}$, while the MBI min scenario results in a total release of 1.42 mol DIP m$^{-2}$ (56 % increase) and the MBI max scenario gives a total release of 1.80 mol DIP m$^{-2}$ (98 % increase). At station E, the total release of DIP is increased by 72 % and 112 % (Table E1).

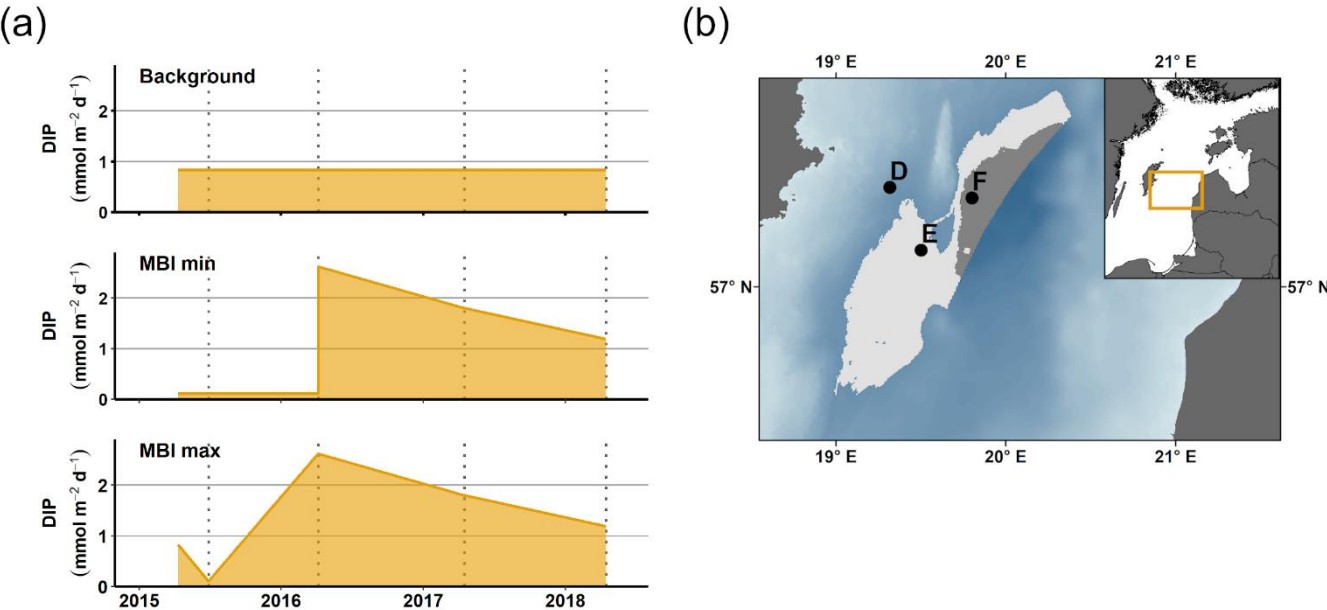

**Figure 5.** Upscaling of DIP release. (a) DIP release at station F according to three different scenarios: Background, MBI min and MBI max. The lines show the DIP flux according to the different scenarios, the shaded area represents the total DIP release during the period. (b) Sediment areas (western flank of the EGB) used for upscaling calculations. The DIP flux per $m^2$ in the light grey area (150-190 m) is assumed to be the same as at station E, while the dark grey area (> 190 m) is represented by station F. Bathymetric data from the Baltic Sea Bathymetry Database (version 0.9.3), the map was created using the software ArcMap$^{TM}$ (v. 10.6) by Esri.

Upscaling of these results (appendix E) further shows the importance of the increased DIP release. The calculation is limited to the western flank of the EGB, since previous studies have shown that the sediment-water fluxes generally are lower on the eastern side (Hall et al., 2017; Nilsson et al., 2019; Sommer et al., 2017; Viktorsson et al., 2013). We assume that the sedimentary DIP release per $m^2$ at 150-190 m depth equalled that at station E, while the release at sediments deeper than 190 m were represented by station F (Fig. 5b). Since measurements are lacking from station D in 2015, only stations E and F are included in the upscaling. Since MBI had a relatively small effect at the fluxes at station D, however, these calculations should still give an acceptable estimate of the total DIP release. The MBI scenarios predict an extra release of 9.57-15.39 kton y$^{-1}$. These numbers can be put in relation to the total release of DIP from oxygen-depleted sediments in the central Baltic Sea (Baltic Proper), which has been estimated to $60 - 150$ kton y$^{-1}$ (Almroth-Rosell et al., 2015; Hall et al., 2017; Sommer et al., 2017; Viktorsson et al., 2013). Even our conservative estimates suggest a substantial increase in sedimentary DIP release in the area following the MBI.

The sedimentary release of DIP led to a steady increase of the deep water DIP concentration in the years following the MBI (Fig. A8; SMHI, 2021). Yet the effect of the excess DIP release on an ecosystem scale depends on the cause of the elevated benthic fluxes. If the MBI enhanced the degradation of the organic matter already present in the sediment, the excess DIP release would increase the water column DIP pool. As discussed in section 3.2, however, the sedimentary $P_{org}$ pools were not

large enough to support the in situ measured fluxes. Enhanced degradation of organic matter due to oxygenation is thus unlikely to have been the main cause of the elevated benthic fluxes. The data instead suggested that the primary driver behind the benthic fluxes was input of organic matter. If the MBI enhanced the organic matter input by accelerating vertical and horizontal shuttling, material would have been rearranged in the basin. The excess DIP release would then not have increased the DIP

pool in the water column, since the same material would have been degraded elsewhere in the basin had the MBI not occurred. However, as discussed above, the events following the MBI (transient oxygenation, resuspension caused by the gravity current) could also stimulate the degradation of materials that was already present in the sediment. Hence, even if most of the excess DIP release was caused by accelerated shuttling, the MBI may have led to an increase in the total DIP release (and the water column DIP inventory) on a basin-wide scale over a three year period.

## 4 Conclusions

In this paper, we show that an oxygenated inflow to long-term anoxic sediments temporarily elevated organic matter degradation, thereby increasing the sedimentary release of DIP. Oxygen is generally believed to be able to break the positive feedback between anoxia and eutrophication (Ruttenberg, 2003; Stigebrandt, 2018), but that was not the case with the short and moderate oxygenation in this study. Instead, our results suggests that the inflow accelerated vertical and horizontal

shuttling of particulate matter which, combined with potential stimulation of organic matter breakdown in the sediment, increased the sedimentary nutrient and DIC effluxes. The enhanced mineralisation of organic matter after oxygenated inflows observed in this study is a previously unrecognised feedback mechanism in eutrophic coastal systems (Breitburg et al., 2018; Middelburg and Levin, 2009).

In estuaries and coastal seas around the world, climate change and land use are predicted to both strengthen and expand oxygen-

depleted zones (Altieri and Gedan, 2015). While oxygenated inflows have been thought to mitigate eutrophication feedbacks by enhancing P burial, our results show that they also can lead to increased mineralisation of organic matter. In long-term anoxic areas, irregular inflows might not bring enough oxygen to oxidise reduced chemical species and promote removal of nutrients. Instead, inflows may cause an internal redistribution of organic matter, leading to a temporal stimulation of organic matter degradation in the sediment. This stimulation of benthic nutrient recycling can lead to increased availability of P and

other nutrients, thereby strengthening eutrophication.

**Table A1.** Description of stations. Bottom water salinity, temperature and oxygen concentration as measured by the benthic chamber lander or CTD. Limit of detection for oxygen was 0.5 µM.

| Station | Depth | Year | Salinity | Temp. (°C) | Oxygen (µM) |
|---------|-------|------|----------|------------|-------------|
| **D** | 130 m | 2016 | 12.5-12.7 | 6.5-6.6 | <0.5 |
|  |  | 2017 | 12.7 | 6.7-6.8 | 8-13 |
|  |  | 2018 | 12.4-12.6 | 6.6-6.9 | <0.5 |
| **E** | 170 m | 2016 | 12.9-13.1 | 6.8-6.9 | 7-20 |
|  |  | 2017 | 13.1 | 6.9-7.2 | 3-13 |
|  |  | 2018 | 13.1 | 6.8-7.1 | <0.5 |
| **F** | 210 m | 2016 | 13.8-13.9 | 7.3-7.4 | 22-30 |
|  |  | 2017 | 13.5 | 7.2-7.4 | <0.5 |
|  |  | 2018 | 13.1-13.3 | 6.9-7.15 | <0.5 |


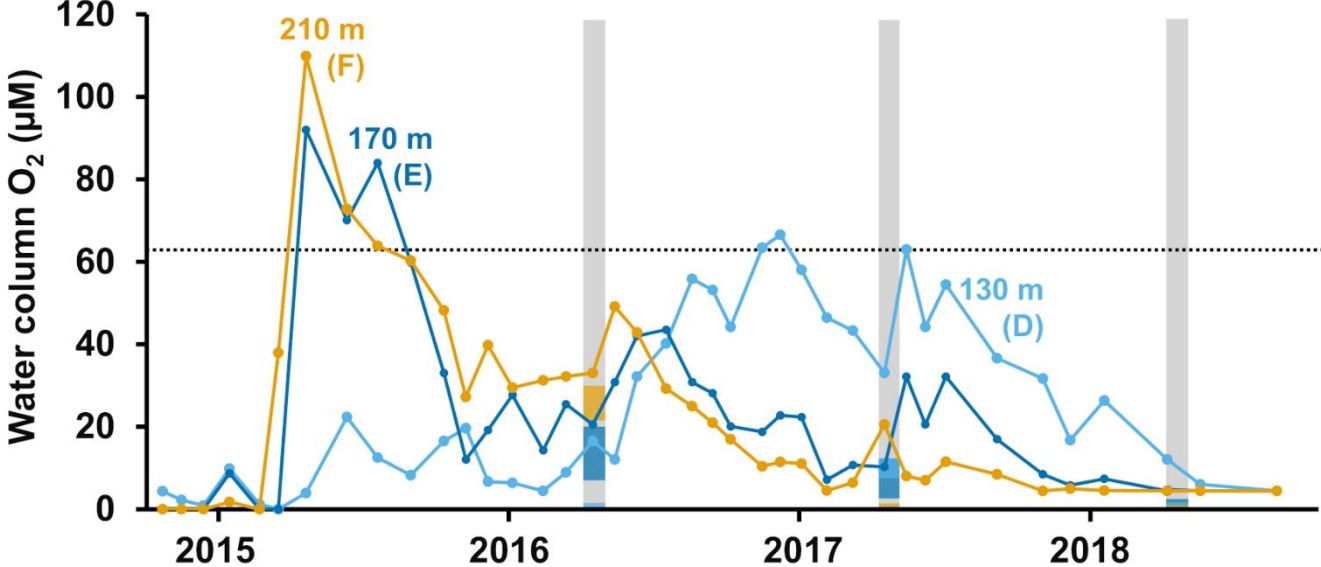

**Figure A1.** Oxygen monitoring data. Water column oxygen concentrations at depths corresponding to stations D-F at the SMHI (Swedish Meteorological and Hydrological Institute) monitoring station BY15 (SMHI, 2021). The grey lines indicate when samplings for this study 470    were conducted, the dotted line shows the limit for hypoxia (63 µM) Colours on the lines show the bottom water concentration measured by the benthic chamber lander at each station. After van de Velde et al. (2020b)

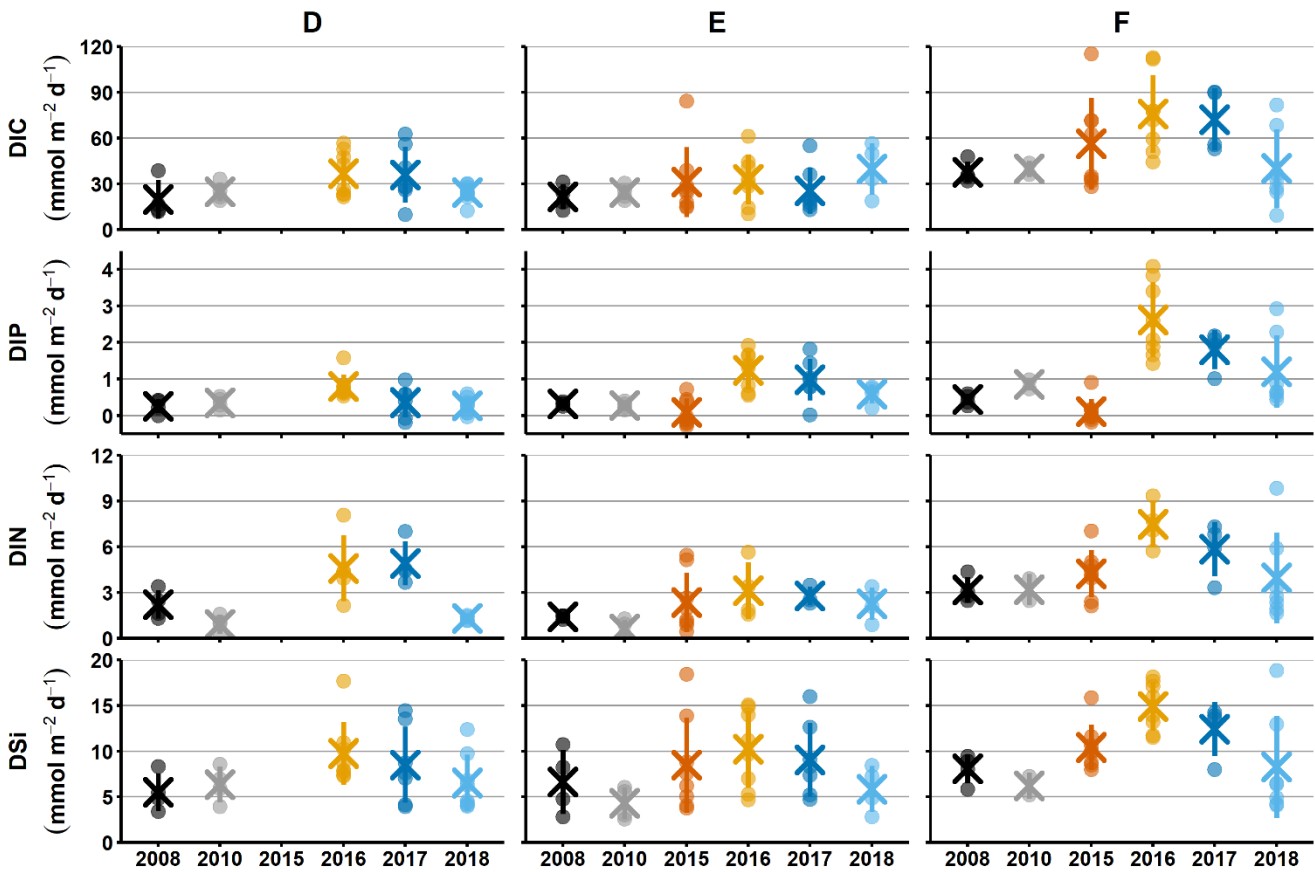

**Figure A2.** Sediment-water fluxes measured with the benthic chamber lander. Flux data are from before the inflow (2008, 2010; Hall et al., 2017; Nilsson et al., 2019; Viktorsson et al., 2013), right after the inflow (2015; Hall et al., 2017) and post inflow (2016-2018; this study). There are no data from station D in 2015.

**Table A2.** Results from the type III ANOVA of sediment-water solute fluxes. Df values for DIN within brackets.

| | Df | DIC | | | | DIP | | | |
|---|---|---|---|---|---|---|---|---|---|
| | | Sum Sq | Mean Sq | F value | p value | Sum Sq | Mean Sq | F value | p value |
| **Year** | 5 | 3.46 | 0.69 | 1.39 | 0.313 | 5.32 | 1.06 | 11.61 | *0.001* |
| **Station** | 2 | 8.66 | 4.33 | 8.72 | *0.008* | 2.69 | 1.35 | 14.68 | *0.001* |
| **Year*Station** | 9 | 2.49 | 0.28 | 0.56 | 0.802 | 0.80 | 0.09 | 0.97 | 0.515 |
| **Deployment (Year, Station)** | 9 | 4.47 | 0.50 | 2.52 | *0.014* | 0.83 | 0.09 | 2.26 | *0.027* |
| **Residual** | 73 | 14.38 | 0.20 | | | 2.960 | 0.041 | | |
| | Df | DIN | | | | DSi | | | |
| | | Sum Sq | Mean Sq | F value | p value | Sum Sq | Mean Sq | F value | p value |
| **Year** | 5 | 25.97 | 5.20 | 6.63 | *0.045* | 6.27 | 1.25 | 3.10 | 0.067 |
| **Station** | 2 | 12.00 | 6.00 | 7.65 | *0.043* | 2.48 | 1.24 | 3.07 | 0.096 |
| **Year*Station** | 9 | 4.18 | 0.46 | 0.59 | 0.764 | 0.60 | 0.07 | 0.17 | 0.993 |
| **Deployment (Year, Station)** | 9 (4) | 3.14 | 0.78 | 1.91 | 0.121 | 3.64 | 0.40 | 2.72 | *0.009* |
| **Residual** | 73 (56) | 22.99 | 0.41 | | | | | | |




**Table A3.** Results from Student-Newman-Keuls (SNK) tests. For years or stations with different group letters, the average fluxes differed significantly. Fluxes were highest in groups named a, with decreasing fluxes in groups following alphabetical order.

| Solute | Year | Groups | Station | Groups |
|--------|------|--------|---------|--------|
| **DIP** | 2008 | c | D | b |
| | 2010 | c | E | b |
| | 2015 | d | F | a |
| | 2016 | a | | |
| | 2017 | b | | |
| | 2018 | bc | | |
| **DIN** | 2008 | b | D | b |
| | 2010 | c | E | b |
| | 2015 | ab | F | a |
| | 2016 | a | | |
| | 2017 | a | | |
| | 2018 | ab | | |
| **DIC** | | | D | b |
| | | | E | b |
| | | | F | a |


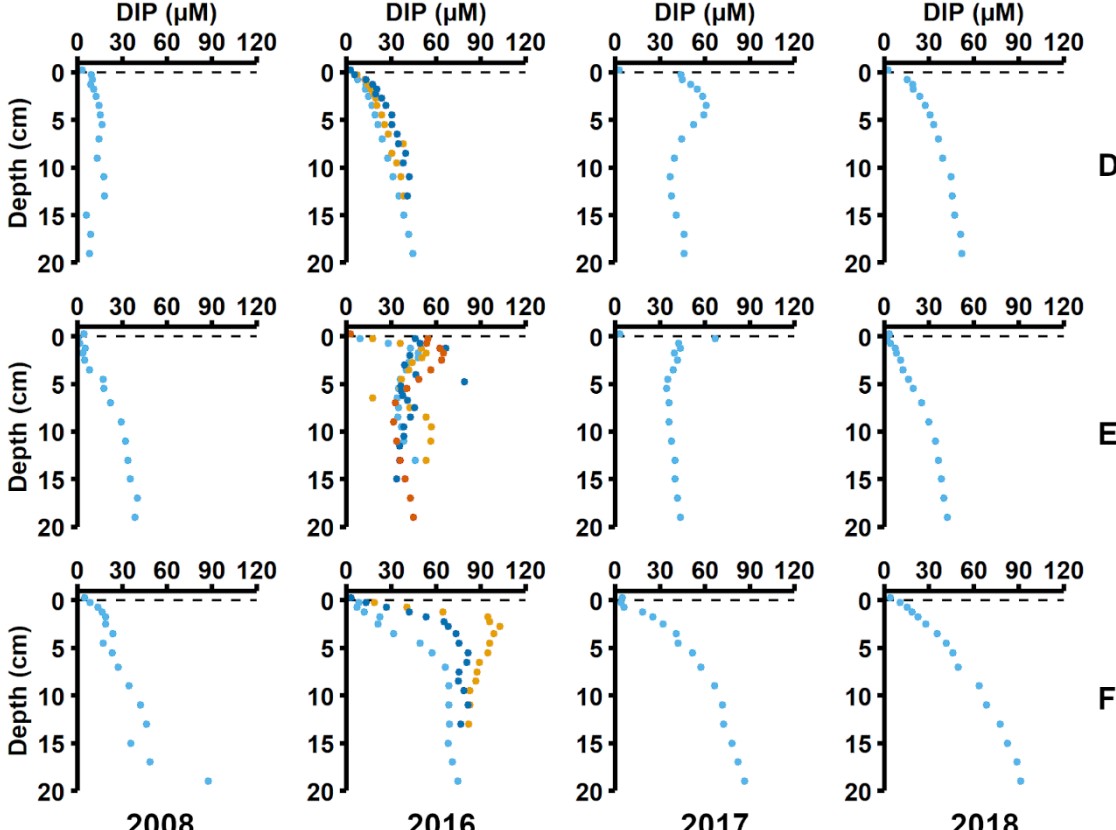

**Figure A3.** Pore-water profiles of dissolved inorganic phosphorus (DIP). Profiles from 2008 are from Viktorsson et al. (2013). Different colours show replicate cores (same as Table C1): light blue markers – core 1, orange markers – core 2, dark blue markers – core 3, red markers – core 4.


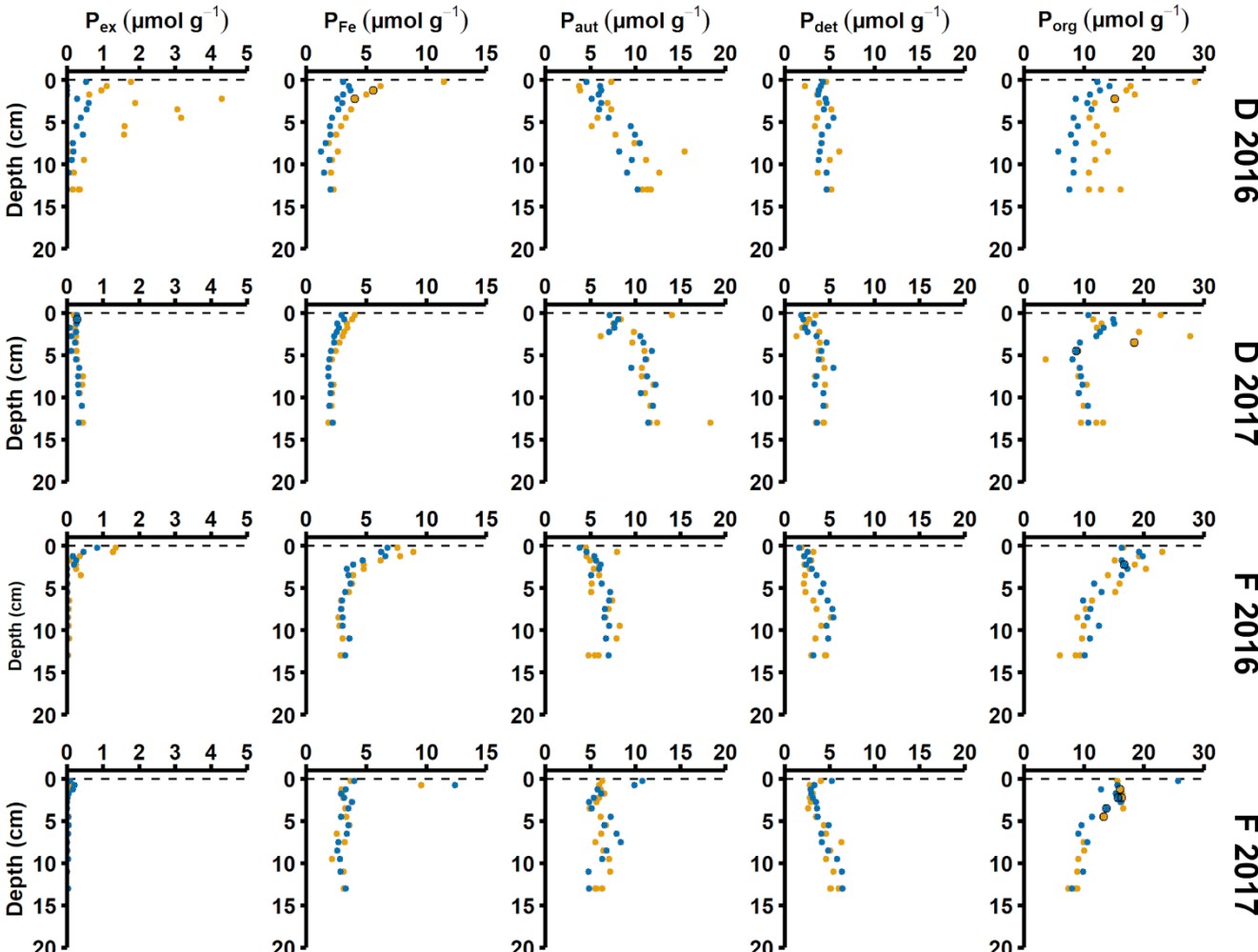

**Figure A4.** Solid phase phosphorus speciation. $P_{ex}$ – exchangeable or loosely sorbed P, $P_{Fe}$ – easily reduced or reactive iron bound P, $P_{aut}$ – authigenic P, $P_{det}$ – detrital P, $P_{org}$ – P associated with organic matter. The blue and orange markers represent replicate small sediment cores (inner diameter 6 cm).

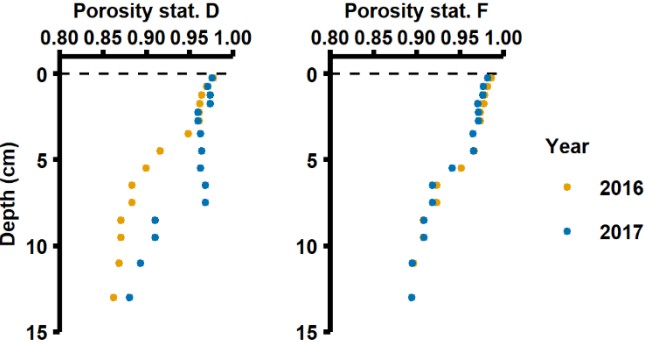

**Figure A5.** Porosity profiles from stations D and F in 2016 and 2017.

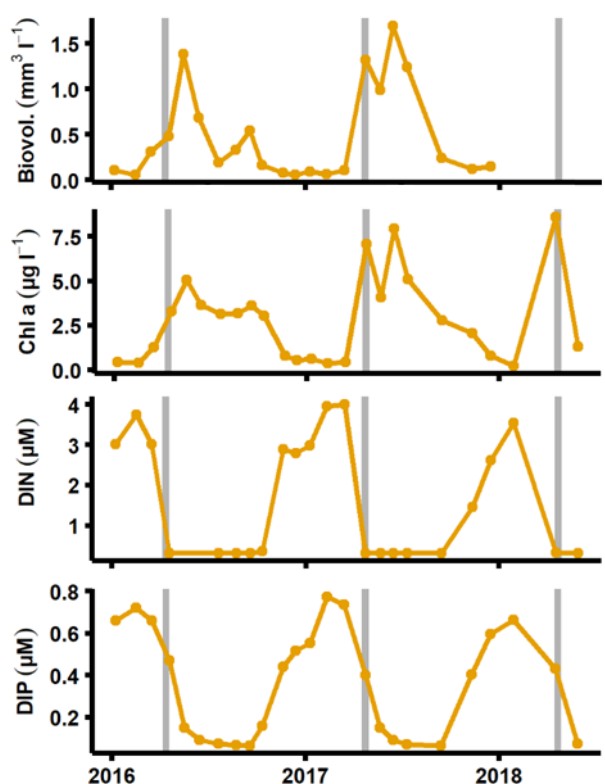

**Figure A6.** Primary production and nutrient monitoring data. Data from SMHI station BY15, top 10 m of the water column (SMHI, 2021). The graphs show (from top to bottom) phytoplankton biovolume, chlorophyll a concentrations, DIN (nitrate + nitrite + ammonium) concentrations and dissolved inorganic phosphorus (DIP) concentrations. The grey lines indicate when samplings for this study were conducted.

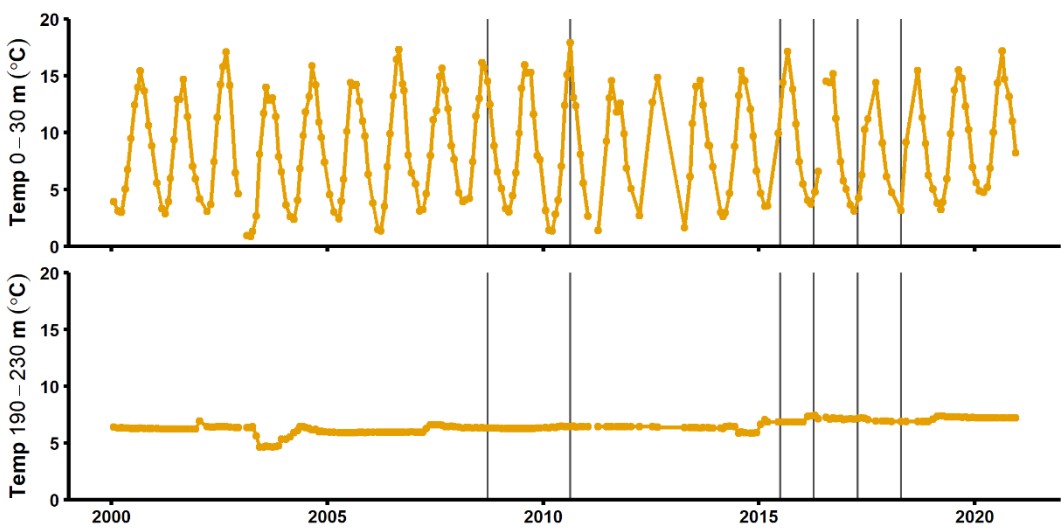

**Figure A7.** Temperature in the surface layer (top) and deep layer (bottom) at the SMHI station BY15 (SMHI, 2021). The grey lines indicate when samplings for this study were conducted (2008, 2010: August/September; 2015: early July; 2016-2018: April).

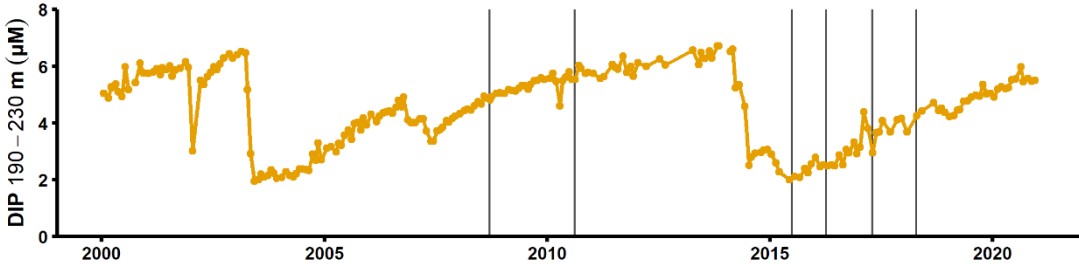


**Figure A8.** Dissolved inorganic phosphorus concentrations in the deep layer the SMHI station BY15 (SMHI, 2021). The grey lines indicate when samplings for this study were conducted (2008, 2010: August/September; 2015: early July; 2016-2018: April).


**Appendix B: Calculation of solid phosphorus inventories**

The solid phase inventory between the sediment surface and sediment depth z was calculated by multiplying the solid phase
density ($\sigma_{sed}$) with the depth-integrated porosity ($\phi$) and concentration of the P fraction ($C_P$) of interest:

$$\text{Inventory} = \sigma_{sed} \int_0^{Zend} \left(1 - \varphi(z)\right) C_P(z) dz \tag{B1}$$

The value for $\sigma_{sed}$ (1.9 g cm$^{-3}$) was taken from van de Velde et al. (2020b). In a few cases (see Figs. 4a and 5a), P concentrations
from certain sediment depths were missing. To not underestimate the sedimentary inventories, the missing values were
replaced by average concentrations calculated from the sediment depths just above and below.

We were interested in the 'excess' inventories of $P_{ex}$ and $P_{Fe}$ (Fig. B1), i.e. is the transiently increased P fraction. For $P_{ex}$, we
consider the excess P inventory to be the inventory as calculated by equation 1, as the concentration of $P_{ex}$ decreased to 0
below 5 cm in both 2016 and 2017. The $P_{Fe}$ profile had a background concentration of ~3 µmol g$^{-1}$ below 5 cm, which was
consistent between sampling years. The concentration at the water interface in 2017 (i.e. the concentration in the settling
particles that had not been exposed to the oxygen-rich water of the MBI) was also close to 3 µmol g$^{-1}$. Therefore, we determined
the 'excess' $P_{Fe}$ concentration as the inventory as calculated by equation 1, subtracted by a background inventory calculated
by replacing $C_P$ in equation 1 with the average concentration below 5 cm.


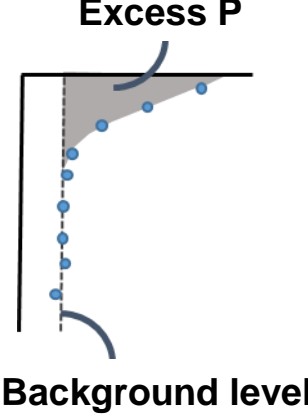

**Figure B1.** Definition of 'excess' phosphorus (P). Schematic figure showing the definition of 'excess' phosphorus (P), which is the integrated
enrichment of a P fraction compared to its background concentration.


## Appendix C: Calculation of diffusive fluxes

Diffusive fluxes (J) were calculated from the pore water profiles of DIP (Fig. A3) according to Eq C1:

$$J = -\varphi D_S \frac{\partial C}{\partial z}$$ (C1)


where $D_S$ is the sediment diffusion coefficient and C is the concentration (Boudreau, 1997). $D_S$ was calculated according to Eq. C2:

$$D_S = \frac{D}{\theta^2} = \frac{D}{1 - \ln(\varphi^2)}$$ (C2)


where D is the diffusion coefficient in free solution and $\theta$ is the tortuosity (Boudreau, 1997). The R package 'marelac' (Soetaert et al., 2020) was used to calculate D.

**Table C1.** Diffusive fluxes of dissolved inorganic phosphorus. The fluxes are calculated from pore water profiles, values are given in mmol m$^{-2}$ d$^{-1}$. The 2008 values are calculated from profiles from Viktorsson et al. (2013). When there are results from multiple cores, the core number is given in brackets. For profiles that displayed production peaks, both the upward flux ($J_{up}$) and the downward flux ($J_{down}$) from that peak are presented.

|   | 2008 | 2016 | 2017 | 2018 |
|---|---|---|---|---|
| **D** | $J_{up} = 0.023$ | $J_{up} = 0.015$ | $J_{up} = 0.276$ | $J_{up} = 0.029$ |
|   | $J_{down} = 0$ | $J_{down} = 0$ | $J_{down} = 0.016$ | $J_{down} = 0$ |
|   | $J_{sum} = 0.023$ | $J_{sum} = 0.015$ | $J_{sum} = 0.292$ | $J_{sum} = 0.029$ |
| **E** | $J_{up} = 0.008$ | $J_{up} = 0.100$ *(1)*, $0.115$ *(2)*, $0.311$ *(3)*, $0.373$ *(4)* | $J_{up} = 0.459$ | $J_{up} = 0.012$ |
|   | $J_{down} = 0$ | $J_{down} = 0.047$ *(1)*, $0.034$ *(2)*, $0.007$ *(3)*, $0.024$ *(4)* | $J_{down} = 0.178$ | $J_{down} = 0$ |
|   | $J_{sum} = 0.008$ | $J_{sum} = 0.147$ *(1)*, $0.149$, $0.318$, $0.397$ | $J_{sum} = 0.636$ | $J_{sum} = 0.012$ |
| **F** | $J_{up} = 0.026$ | $J_{up} = 0.165$ *(1)*, $0.092$ *(2)*, $0.030$ *(3)* | $J_{up} = 0.053$ | $J_{up} = 0.041$ |
|   | $J_{down} = 0$ | $J_{down} = 0.010$ *(1)*, $0$ *(2)*, $0$ *(3)* | $J_{down} = 0$ | $J_{down} = 0$ |
|   | $J_{sum} = 0.026$ | $J_{sum} = 0.175$ *(1)*, $0.092$ *(2)*, $0.030$ *(3)* | $J_{sum} = 0.053$ | $J_{sum} = 0.041$ |

**Appendix D: Pigment extractions**

Pigment samples were collected in 2008, 2017 and 2018. Sediment cores were collected with a multiple corer and were sliced
at 0.5 cm resolution from 0 to 2 cm depth. From each slice, 1 mL of sediment was transferred to 15 mL centrifuge tubes and
was subsequently frozen in liquid nitrogen. The samples were extracted in 10 mL of an acetone:methanol (80:20) solution at -
20°C for 24 h before they were sonicated for 60 s and filtered through 0.45 μm nylon filters. Extracts were separated and
identified by high performance liquid chromatography (HPCL, Shimadzu LCsolutions System) with photodiode array
detection (SPD-M20A). The relative concentration of sample chlorophyll a was determined from calibration curves acquired
from analysis of an external standard (DHI LAB).

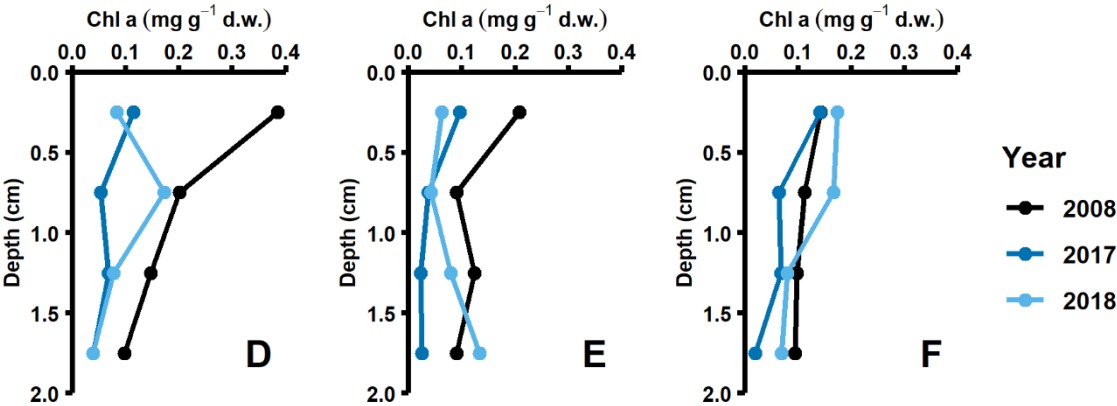

**Figure D1.** Chlorophyll a profiles. Profiles of chlorophyll a (Chl a) in the top 2 cm of the sediment.

## Appendix E: Upscaling calculations

Bathymetry data from the Baltic Sea Bathymetry Database (version 0.9.3) were used for the upscaling calculations. Sediment surface areas were extracted using the tool 'Surface Volume' in the software ArcMap[TM] (v. 10.6) from Esri. The area at 150-190 m depth was represented by station E, while sediments deeper than 190 m were represented by station F. The extracted areas were multiplied with the total DIP release calculated for the Background, MBI min and MBI max scenarios, and were then divided by the time used in the total release scenarios (3.13 years) to obtain an areal release per year.

**Table E1.** Total release of dissolved inorganic phosphorus (DIP). The total DIP release represent the entire time period from the arrival of oxygen until the end of the study (3.13 years). The total DIP release was not calculated for station D since no flux measurements were conducted at the site in 2015.

| Total DIP release | | | |
|---|---|---|---|
| | Background (mol m$^{-2}$) | Min release (mol m$^{-2}$) | Max release (mol m$^{-2}$) |
| **Station E** | 0.43 | 0.74 | 0.91 |
| **Station F** | 0.91 | 1.42 | 1.79 |

| Upscaling | | | |
|---|---|---|---|
| | Background (kton y$^{-1}$) | Min release (kton y$^{-1}$) | Max release (kton y$^{-1}$) |
| **150 – 190 m** (2179.518 km$^2$) | 9.27 | 15.96 | 19.62 |
| **> 190 m** (571.7793 km$^2$) | 5.15 | 8.03 | 10.19 |
| **Sum** | 14.42 | 23.99 | 29.81 |
| **Excess release** | | 9.57 | 15.39 |







*Data availability* Data presented in this manuscript is available from the VLIZ data repository (https://doi.org/10.14284/442).

*Author contribution* AH, SV, MK, EAR and PH designed the study. AH, SV, EAR and PH conducted sediment sampling, MK coordinated lander deployments. AH, SV and ML carried out the SEDEX. AH and SV performed calculations, interpreted the data and wrote the manuscript with input from all authors.

*Competing interests* The authors declare that they have no conflict of interest.

*Acknowledgements* We thank the crew of the University of Gothenburg R/V *Skagerak* for support at sea, Martine Leermakers for assistance with the ICP-OES analysis, Per Bergström for statistical help and Elizabeth Robertson for proofreading the manuscript.

*Financial support* This work was supported by the Swedish Research Council (VR grant 2015-03717 to PH). SV was supported by an appointment to the NASA Postdoctoral Program at the University of California, Riverside, administered by Universities Space Research Association under contract with NASA and is currently supported by by the Belgian Science Policy office (FED-tWIN2019-prf-008 – ReCAP project grant).

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
