# Peer review of "Deep-water inflow event increases sedimentary phosphorus release on a multi-year scale"

_Biogeosciences, 2021_

## Author Comment (AC1)

Responses to comments on

**"Deep-water inflow event increases sedimentary phosphorus release on a multi-year scale"**

We thank the reviewers for positive and constructive feedback. We have carefully considered and addressed all comments, which greatly improved the manuscript. In additions to linguistic corrections and smaller modifications described in the replies below, we have made some more substantial changes. In brief:

- We have calculated the sedimentary inventories of $P_{org}$ and compare these to the measured fluxes. Based on this comparison (and previous studies), we argue that an increased input of organic matter is the most likely explanation for the enhanced benthic fluxes. The implications of this conclusion on a system level are discussed.

- The calculations of the $P_{ex}$ and $P_{Fe}$ inventories have been revised. Most notably, a conversion error led to underestimated inventories at station D in the original manuscript. However, the updated calculations do not change any trends or conclusions.

- The benthic flux evaluation has been revised following feedback received while this manuscript was under review. Most notably, after visual inspection of all data, we have chosen not to exclude fluxes for which the slope of the regression line has a p-value > 0.05. These low, "non-significant" fluxes only comprise 5 % of all flux measurements, but are retained to avoid overestimated average fluxes. The updated fluxes are generally slightly lower, but do not change any trends or conclusions.

Our replies to the reviewers' comments are outlined below in blue.

**Reviewer 1**

**General comments**

Sedimentary phosphorus, which can be remobilized by diagenetic processes and reach the water column, thus increasing the risk of eutrophication, is essentially found in two reactive forms, P associated with organic matter and P associated with iron oxides. Iron oxide-bound P is released when iron oxides are reduced and the iron is not rapidly reoxidized. This occurs in sediments that come into contact with anoxic bottom water, or at least, free of oxidants for Fe(II), bearing in mind that dissolved oxygen is not the only possible oxidant. The P associated with organic matter is released when the organic matter is mineralized. The rate of OM mineralization depends on several factors. The presence of strong oxidants such as oxygen or nitrate promotes efficient mineralization.

In the study presented here, the authors emphasize a counter-intuitive fact, but one that is based on correct reasoning. Indeed, the authors describe the evolution of benthic P fluxes in relation to the oxygen concentration in bottom waters of areas of the Baltic Sea that have been studied for a long time. It is known that the anoxic periods that characterize the bottom of the Baltic Sea favour benthic P fluxes. In some years, oxygenated waters from the North Sea invade the Baltic Sea bottom. The expected effect is a trapping of P due in particular to iron oxides which become stable, and thus a decrease in the benthic fluxes of P. However, the authors show by flux monitoring data following an oxygen supply episode, that the fluxes of P have increased. They explain this by the fact that the oxygen supply significantly increases the rate of OM mineralization and that it is this process that increases the P fluxes. This is supported by dissolved inorganic carbon (DIC) flux data that point in this direction.

The conclusions are important, the idea is elegant. However, I would suggest that the authors provide some clarification on elements for discussion, and some clarification on the presentation of the results.

We thank the reviewer for thorough and constructive comments.

**specific comments**

1) When the bottom water becomes oxic, only a very fine fringe of sediment also becomes oxidized. Most of the sediment column (probably here from the 1st cm below the water-sediment interface) is anoxic and does not change its redox state when the supernatant water undergoes redox oscillations. The oxides present in the fine oxidized fringe of the sediment most probably have their adsorption sites very quickly saturated and do not trap the P that diffuses from the sediment. This nuance should be mentioned.

It is indeed likely that there was a rapid saturation of metal oxides by DIP since the oxygenation was restricted to the most surficial part of the sediment. We have added a paragraph discussing this:

*"Following the MBI, there was a substantial elevation of the sedimentary DIP release in the EGB despite oxygenation of the sediment. Although Fe and Mn oxides were present at the sediment surface (Fig. 3a, Fig. 4a; Dellwig et al., 2018; Hermans et al., 2019b), they did not prevent DIP from escaping to the water column. The oxygen penetration was restricted to the top 2 mm of the sediment, and fluxes of hydrogen sulphide and other reduced compounds from deeper sediment layers appear to have prevented extensive oxidation of the sediment (Hermans et al., 2019b;*

*Marzocchi et al., 2018). In the eastern part of the EGB, the ratio between Fe oxides and $P_{ex} + P_{Fe}$ and was ~1 (Hermans et al., 2019b), suggesting that the adsorption sites on the Fe oxides were saturated since the lowest ratio Fe oxide:P ratio that has been observed in marine environments is ~2 (Gunnars et al., 2002). Other minerals, such as Mn oxides and carbonates, could also act as sorption sites for phosphate (Kraal et al., 2017; Yao and Millero, 1996). After the MBI, large quantities of Mn oxides were deposited on the sediment (Dellwig et al., 2018; Hermans et al., 2019b; van de Velde et al., 2020b). Hermans et al. (2019b) found clear associations between Mn and P enrichments in the sediment, and stipulated that Mn minerals had an important function in the retention of phosphate, which could explain the relatively large amount of $P_{ex} + P_{Fe}$ relative to Fe oxides. Nevertheless, the fact that DIP escaped from the sediment to the water suggests either that all adsorption sites were saturated, or that the adsorption process was not efficient (e.g. due to kinetic limitations) relative to the diffusion of DIP toward the water column. Overall, it appears that the sediment of our study site had limited capacity to trap DIP during this short oxygenation event."*

2) Fig. 1 : the 'viscious cycle': The diagram shows P entering the cycle, but no exit route for this P. In this cycle, it seems that 100% of the P that reaches the sediment is then recycled. Shouldn't we consider a fraction that remains in the sediment in an authentic form for example?

It is correct that a fraction of the DIP is buried, and some DIP is also transported to the North Sea and the Bothnian Sea. However, we have opted to remove the figure, since (as pointed out by reviewer 3) the mechanisms behind the 'vicious cycle' are described in the text and have been discussed thoroughly elsewhere.

3) line 47: "Fe and Mn oxides adsorb DIP" OK for Fe oxides. It is less obvious for Mn oxides.

It has been shown that DIP is adsorbed onto Mn oxides and that they can have a significant effect on the P retention in systems where the concentration of DIP is much higher than the available surface sites of iron oxides (e.g. low-oxygen areas in the Baltic Sea) (Yao and Millero, 1996). We have modified the introduction, however:

*"Firstly, iron (Fe) and (to some extent) manganese (Mn) oxides, which adsorb and retain DIP in the sediment under oxic conditions, are reduced and solubilised in anoxic environments (Jilbert and Slomp, 2013; Yao and Millero, 1996)."*

Further discussion about the role of manganese oxides in sedimentary DIP retention is described in the reply to question 1.

4) section 2.1 or section 3: nothing is said about the nature of the sediments: grain size, POC content, porosity... This is general information necessary for a better understanding of the system (and for checking the inventory calculations).

We have added information about POC concentrations and porosity to section 2.1 and porosity profiles have been added to figure A4.

*"The sediment is rich in organic matter (organic carbon 10-15 % dry weight) and has a high surface porosity (0.95-0.98; Nilsson et al., 2019; van de Velde et al., 2020b)."*

5) line 131 "The sediment was transferred into centrifuge tubes and pore water was collected using Rhizon samplers" I don't understand: the interstitial waters were collected by centrifugation or with rhizon? or both?

Rhizon samplers were inserted into the sediment contained in the centrifuge tubes; this has been clarified in the text.
*"The sediment was transferred into centrifuge tubes into which Rhizon samplers (Rhizosphere research products) were inserted for collection of pore water."*

6) line 177: the authors refer to dissolved silica data which are unfortunately not shown here. This is unfortunate, because the increase in Si fluxes could be an indicator of the increase in bioturbation which is only briefly discussed here. The article cited below co-authored by one of the co-authors of the present manuscript is probably the best example of this: van der Loeff, M. M. R., Anderson, L. G., Hall, P. O. J., Iverfeldt, Å., Josefson, A. B., Sundby, B., & Westerlund, S. F. G. (1984). The asphyxiation technique: An approach to distinguishing between molecular diffusion and biologically mediated transport at the sediment—water interface. Limnology and Oceanography, 29(4), 675-686. doi:10.4319/lo.1984.29.4.0675

Please see the reply to question 9.

7) Figure 3 and the other figures: the shades of colour to represent the different sampling periods are not contrasted enough. An effort must be made by the reader to locate the dates. The quality of the figures needs to be improved.

We have changed the colour schemes in the figures to increase the contrast.

8) line 2015-225: Here is a major point; The authors compare the fluxes with the inventories of P present in the sediment. They show that the inventory of P associated with iron oxides cannot explain the fluxes. However, since the authors hypothesize an increase in benthic P fluxes related to OM mineralization, it is interesting to compare these fluxes with the inventory of P associated with OM. I took the liberty of doing so based on the OM-associated P content of Figure A4 (about 15 µmol/g) and a porosity of 0.8. I find that the inventory of P associated with OM is 400 mmol/m2 for a sediment thickness of 5 cm. The measured flux of P is 2 mmol/m2/day, which means that this flux would correspond to a total mineralization of P associated with organic matter (and thus OM) in 200 days over a thickness of 5 cm. This is not consistent with the data and poses a major problem with regard to the representativeness of the measured fluxes. The measured P flux of 2 mmol/m2/day gives the impression of extracting all the P from the sediment. This is not the case if, in addition to the outgoing fluxes, there are equivalent incoming fluxes via sedimentation, which must then be described. This quantitative control is necessary.

This is indeed an important point. We have calculated the Porg inventory in the top 5 cm of the sediment and, as suggested by both reviewers 1 and 2, and added a paragraph discussing the in situ measured fluxes in relation to the Porg inventories and diffusive fluxes calculated from pore water profiles. As suggested by reviewer 2 and previous studies, the main reason for the discrepancy between the lander-measured fluxes and diffusive fluxes/inventories calculated from sediment-core profiles is likely that organic matter was degraded rapidly at the sediment-water interface. In addition to the vertical input of phytoplankton material, there is a substantial and continuous horizontal shuttling of material from shallow to deep parts of the Eastern Gotland Basin.

*"The in situ measured DIP fluxes were considerably higher than diffusive fluxes calculated from pore-water profiles (Table C1). Moreover, the $P_{org}$ inventories in the top 5 cm of the sediment (Fig. 3b, Fig. 4b) could only support the DIP fluxes measured in 2016-2017 for 59-159 days at station D and 15-24 days at station F. This discrepancy between measured fluxes and calculated diffusive fluxes and sedimentary $P_{org}$ inventories may partly be explained by potential overestimation of measured fluxes (see section 2.2). However, even in the unlikely case that the measured fluxes overestimated the in situ fluxes by 50 %, the $P_{org}$ inventories would only support the DIP fluxes for 118-318 days at station D and 30-48 days at station F. Alternatively, it is well known that chemical gradients at the sediment-water interface may not be accurately captured by the limited vertical resolution of pore water profiles, resulting in underestimates when calculating diffusive fluxes (Nilsson et al., 2019; Noffke et al., 2012; Sundby et al., 1986; van de Velde et al., 2020b). Since the sediment at stations D and F had such high porosities (~0.98; van de Velde et al., 2020b; Fig A5), it is also possible that the topmost fluffy and organic rich layer was not captured during sampling and processing of sediment cores. Solute fluxes caused by processes in this sediment layer would have been detected in the benthic chamber lander incubations, however. The high in situ measured fluxes relative to calculated diffusive fluxes and sediment $P_{org}$ inventories thus suggests that processes occurring at the sediment-water interface were the main driver behind the benthic fluxes, whereas sedimentary inventories played a minor role. Previous studies have indeed suggested that organic matter mineralisation is rapid and efficient in the deep basins of the EGB (Nilsson, 2018; Nilsson et al., 2021), indicating that it mostly takes place at the sediment surface. In addition to vertical deposition of phytoplankton material, there is an extensive horizontal shuttling of material from shallow to deeper parts of the EGB (Christiansen et al., 2002; Jonsson et al., 1990; Leipe et al., 2011) which is thought to support about half of the benthic organic matter mineralisation in the deep basin on an annual basis (Nilsson et al., 2021).*

*If the topmost sediment layer was not accurately captured during sampling, the calculated $P_{ex}$ and $P_{Fe}$ inventories are likely also too low. This does not necessarily mean that the change in inventories between years is underestimated, if equally large fractions of $P_{ex}$ and $P_{Fe}$ were missed both years. Much of the change in $P_{ex}$ and $P_{Fe}$ between 2016 and 2017 is expected to have taken place at the sediment-water interface, however, suggesting that the inventory change was in fact larger than estimated here. Nevertheless, the constant DIC:DIP flux ratio in all years but 2015, together with similar flux patterns for multiple biogenic solutes, suggest that the contribution from $P_{ex}$ and $P_{Fe}$ to the total DIP flux was, in fact, small."*

These results imply that an increased input of organic matter as a result of the MBI is the most likely reason for the elevated benthic fluxes. A paragraph has been added to discuss the implications of this, see reply to reviewer 2.

9) line 291-296: the role of bioturbation is too quickly dismissed here. It is written that there is no sign of colonization by animals. However, the increase of Si fluxs is a possible explanation (see the reference cited above). In lines 302 to 304, an ad hoc explanation is given to describe the Si fluxes (that we would like to see). I think bioturbation is a better explanation. Moreover, the colonization of the sediment by fauna as a result of oxygenation of the waters makes sense and is in line with the authors' conclusions: an increase in mineralization of OM.

We understand the reasoning of the reviewer; however, we disagree with this assessment. The evidence for recolonization of animals during the transient oxygenation is weak. Firstly, the oxygenation lead to bottom-water oxygen concentrations of < 50 µM, which means the water was still hypoxic. It is unlikely that such an environment would have supported high activity of benthic fauna. Secondly, the two specimens of *Bylgides sarsi* mentioned in the manuscript were found during the 2015 expedition described in Hall et al. (2017). In none of the other sediment samples collected (including multiple cores, box cores and sediment retrieved with the lander), at any point during that or the following years, did we observe any animals or signs of animal activity in the laminated sediment. We deem it unlikely that such a low presence of animals that we did not observe them at any occasion could cause this substantial change in benthic fluxes. Furthermore, in a study from the same area and time period by Hermans et al. (2019, "Abundance and Biogeochemical Impact of Cable Bacteria in Baltic Sea Sediments"), they especially searched for animals but could not find any. We have now clarified this reasoning and expanded the discussion about bioturbation:

*"Oxygenation can allow re-colonisation of previously anoxic sediments by benthic animals. Their presence would result in sediment reworking (bio-mixing) and bio-irrigation, which are believed to stimulate organic matter degradation (Ekeroth et al., 2016; van de Velde et al., 2020a). Two individuals of Bylgides sarsi were found near our sampling sites in 2015 (Hall et al., 2017; Stigebrandt et al., 2018). However, all sediment retrieved from the newly oxygenated area during our sampling was laminated and there were no signs of colonisation by animals, consistent with other studies conducted in the EGB in 2016 (Hermans et al., 2019a). It is thus likely that the two specimens of Bylgides sarsi had arrived with the inflowing water (Stigebrandt et al., 2018). The anoxic water layer at 100-140 m depth would further have created a barrier that prevented recolonization of the newly oxygenated sediments. We therefore exclude animal activity as an explanation for the enhanced sediment-water fluxes after the MBI."*

10) Dissolved oxygen concentrations are low, even after North Sea inflow events. The environment is hypoxic. Regarding OM mineralization, oxygen is a powerful oxidant, but so is nitrate. Nitrate is also an oxidant for Fe(II). In this type of eutrophic environment, nitrate concentrations are probably very high. Have they been measured? Do the dynamics of nitrate follow that of oxygen in the chronicle presented here, before, during and after the oxygenation event?

We did study the sedimentary nitrogen cycle during our samplings, but the manuscript describing those data is still in preparation and cannot be cited according to Biogeosciences' guidelines. In the years when there was oxygen in the bottom water, benthic nitrate reduction was indeed active, but at very low rates. The nitrate concentrations in the bottom water reached ~10 µM in 2015 (Hall et al., 2017), but rapidly decreased over the following years in line with the chronicle presented here.

**Reviewer 2 (Tom Jilbert)**

The study "Deep-water inflow event increases sedimentary phosphorus release on a multi-year scale" presents a thought-provoking hypothesis concerning the effect of deep water oxygenation on DIP fluxes from the sediments of the Eastern Gotland Basin (EGB), Baltic Sea. In a nutshell, the authors propose that the repeated inflows of 2014-2017, which led to persistently oxygenated bottom water conditions at their study sites, promoted organic matter remineralization in the surface sediments, leading to an overall enhancement in the flux of DIP to the water column relative to that observed during the typical anoxic conditions. According to the authors' calculations, the post-oxygenation flux of DIP exceeded that predicted from the re-release of P trapped immediately following the initial oxygenation, hence leading to the conclusion that an additional P source (OM remineralization) is required to explain the results. The observations run contrary to a long-held paradigm concerning retention of DIP by sediments following oxygenation. Several mechanisms are proposed to explain the results, including changes in organic matter supply and enhanced rates of remineralization under oxic conditions.

The main body of data supporting the core hypothesis is derived from lander-based flux measurements of DIP, DIC, DIN and DSi. This dataset is a very high-value component of the paper. The level of replication is good, and the parallel trends in the fluxes of all biogenic parameters during the period 2015-2018 appear robust. The more problematic aspects of the study concern the attempted closed-sum budgets of P cycling, including the upscaling of the data to larger areas of the EGB. Some of the interpretations and assumptions here need to be better elucidated. In general I am supportive of the study and especially the robust flux, porewater and sediment data, but I feel that the further treatment of that data needs improvement to be ready for Biogeosciences.

*We are grateful for this in-depth and constructive review.*

Major comments:

As raised by the first reviewer, there is an issue concerning the closed-sum budget of in-situ P pools in the sediments and the estimated DIP fluxes, that requires further elaboration in the text. The range of measured DIP fluxes at station F during 2016-2018 is 2-4 mmol/m2/d (Fig. 3a, b). The first reviewer calculates that a 2 mmol/m2/d flux would exhaust the P-org supply of the upper 5 cm of the sediment column in 200 days, assuming porosity 0.8 and P-org = 15 umol/g. I repeated this calculation with a porosity of 0.9 (probably closer to the true value in the EGB) and of course the duration is even shorter due to the lower volumetric P-org inventory in the sediments (less than 100 days, exact value depends on assumed sediment density). The flux data are correct, and so are the sediment profiles, so to me the only explanation can be that the turnover of biogenic material at the very surface of the sediments (e.g. in the "fluffy layer") must be extremely rapid, therefore dictating the high fluxes. This is very important because it has an implication for the further interpretations. If the observed DIP flux cannot be sustained by the in-situ pool of P-org (as measured in the sediment profiles), it must be sustained by an incoming flux of organic material to the sediment surface. The higher fluxes in 2016-2018 therefore indicate first and foremost that the input of organic material has increased at station F. I suggest that the authors use their own porosity data, if available, to carry out this exercise for themselves and present the results and implications in a revised version.

*This is an important issue. As suggested, we have calculated $P_{org}$ inventories and added a paragraph discussing the in situ measured fluxes in relation to sedimentary inventories and diffusive fluxes calculated from pore water profiles (see reply to question 8 by reviewer 1). We agree with the*

interpretation that the mismatch between fluxes and sedimentary inventories indicate that the fluxes were driven by processes at the sediment-water interface (the "fluffy layer"). This does indeed have implications for the interpretation of the results and we have expanded this discussion (see below).

Following on from this line of thought, I am concerned that the study interprets the enhanced fluxes (and especially DIP fluxes) at these study locations too simply. For example, section 3.4 is devoted to implications of DIP release for coastal deoxygenation and is set up in a way that extrapolates the observed fluxes to a larger area, implying overall enhancement of DIP fluxes and therefore a possible feedback to eutrophication/hypoxia. On the contrary, I would challenge the authors that what is being observed here is an internal rearrangement of the distribution of organic material in the deep areas of the EGB after the inflow, leading to changes in remineralization rates at different locations. Therefore the enhanced fluxes at one location are presumably matched by lower fluxes at another location where the same organic matter would be sedimented under normal conditions. The authors refer to several recent papers describing possible physical mechanisms such as gravity currents and particle aggregation following oxygenation, which could modify the spatial distribution of sedimentation of organic material. I think these are critical to understanding the results. For example, I would ask the question whether we are looking primarily at an enhancement in particle shuttling into deeper areas following the inflow? (it is interesting to note that the deeper site F shows the strongest effects on fluxes, see e.g. Fig. A2, despite a similar oxygenation timeline to the shallower site E, see e.g. Fig. A1). I do not deny that remineralization of OM under oxic conditions can be more rapid (at least for certain compounds; I refer to the cited Arndt et al. 2013 ESR paper for the concept) and hence that the observed effects on fluxes can be influenced directly by the presence of oxygen. However the fact remains that the in-situ pool of OM is insufficient to support the temporally-extrapolated fluxes as presented in Fig. 5, under either oxic or anoxic conditions. So the authors must emphasize the role of variable carbon inputs *(donor control* in the terminology of Arndt et al.) in their interpretation.

We agree with the interpretation that the MBI redistributed sedimentary material, which in turn led to enhanced organic matter mineralization rates in the deeper parts of the EGB. As pointed out, this would not lead to a net increase in the water column DIP pool, just shift where the organic matter degradation takes place. Most likely, however, the enhanced benthic fluxes were caused partly by redistribution of organic matter and partly by mineralization of organic matter that would otherwise have been buried. In that case, there would have been an enhanced release of DIP on the basin scale. We have added text where this is discussed:

*"In contrast, lateral transport could have been a substantial source of POC and associated elements after the inflow. Increased turbidity was measured close to the seafloor below 140 m depth in March 2015, as a gravity current generated by the inflow moved along the slope (Schmale et al., 2016). The inflow itself could thus have enhanced the constantly ongoing lateral transport of relatively fresh organic matter from shallower to deeper areas (Nilsson et al., 2021). Furthermore, the speed of the gravity current generated by the inflow was higher than current speeds generally observed in the area during stagnation periods (Hagen and Feistel, 2004; Schmale et al., 2016). The higher current speed could have resulted in a more intense resuspension and of larger particles (Danielsson et al., 2007), including material that would otherwise have been buried. This mixing of old and new organic matter could additionally have induce a 'priming' effect, where older material is broken down more efficiently in the presence of the fresher material (Bianchi, 2011; van Nugteren et al., 2009)."*

*"The sedimentary release of DIP led to a steady increase of the deep water DIP concentration in the years following the MBI (Fig. A8; SMHI, 2021). Yet the effect of the excess DIP release on an ecosystem scale depends on the cause of the elevated benthic fluxes. If the MBI enhanced the*

*degradation of the organic matter already present in the sediment, the excess DIP release would increase the water column DIP pool. As discussed in section 3.2, however, the sedimentary $P_{org}$ pools were not large enough to support the in situ measured fluxes. Enhanced degradation of organic matter due to oxygenation is thus unlikely to have been the main cause of the elevated benthic fluxes. The data instead suggested that the primary driver behind the benthic fluxes was input of organic matter. If the MBI enhanced the organic matter input by accelerating vertical and horizontal shuttling, material would have been rearranged in the basin. The excess DIP release would then not have increased the DIP pool in the water column, since the same material would have been degraded elsewhere in the basin had the MBI not occurred. However, as discussed above, the events following the MBI (transient oxygenation, resuspension caused by the gravity current) could also stimulate the degradation of materials that was already present in the sediment. Hence, even if most of the excess DIP release was caused by accelerated shuttling, the MBI may have led to an increase in the total DIP release (and the water column DIP inventory) on a basin-wide scale over a three year period.”*

It is of course a fascinating result that there is a genuine increase in the flux of DIP from sediments to the water column in 2016-2018, despite the oxic bottom water conditions. This does indeed run contrary to the often-cited paradigm of P retention with Fe oxides under oxic conditions (e.g. early Einsele and Mortimer papers). This begs the question, why should the P not be retained after release from OM during remineralization? Is it simply a question of the relative availability of Fe-Mn oxides with respect to P, or do the authors have an alternative hypothesis? Note that Fe:P ratios are often used in limnological studies to estimate potential P retention in sediments (Jensen et al Hydrobiol. 235, 1992). If the authors have bulk sediment Fe and Mn contents (or even e.g. CDB-Fe, CDB-Mn data) these could be added to aid this investigation. At the very least, further discussion is required on this important topic.

We do unfortunately not have data of CDB-Fe or CDB-Mn. However, the study by Hermans et al. (2019, “Impact of natural re-oxygenation on the sediment dynamics of manganese, iron and phosphorus in a euxinic Baltic Sea basin”) present detailed data of sedimentary Fe and Mn. As suggested, their study indicate that at least the availability of Fe was low relative to P. We have added a paragraph discussing this, see reply to question 1 by reviewer 1.

Minor comments:

Line 61, 340: The reference to Jilbert and Slomp (2013) is not ideal for this concept. That paper focused more on the role of particle shuttling in stimulating authigenic mineral formation. I suggest to find an alternative.

We have changed the reference to Stigebrandt (2018), “On the response of the Baltic proper to changes of the total phosphorus supply”, Ambio 47:1.

Line 195-200 and Fig 3: It took me a while to understand the message here, which is that an impact of Fe-P dissolution on the flux data would be seen as a DIC:DIP ratio even lower than the mean value of 40 observed in 2016-2018. I think the figure can be annotated to improve this concept. Perhaps add another line for the ratio of 40, or a segment for the ratio 20-70 as referred to in the text, and

modify the text to link better to the figure. Also label the lines in the figure (including the existing Redfield line) with the corresponding ratios.

We have modified the figure; we have added labels to the lines marking the Redfield ratio, and shaded areas marking C:P ratios of 20-70. The paragraph has been partly rewritten to make the message clearer and we explain the causes of the low DIC:DIP flux ratios observed under normal (long-term anoxic) conditions.

*"The decrease in DIP flux at station F in 2015 temporarily raised the DIC:DIP flux ratio above the Redfield ratio (Redfield, 1958) of 106:1 (Fig. 2b; Hall et al., 2017). As the DIC:DIN ratio did not change, the lowered DIP flux was most likely due to trapping of P by Fe and Mn oxides, formed at the sediment surface as a result of the oxygenation. When anoxia returned in 2016, $P_{Fe}$ that had been temporarily trapped during oxic conditions was expected to be released and increase the sedimentary DIP efflux, which would lower the DIC:DIP flux ratio compared to pre-MBI conditions. However, the DIC:DIP ratio only decreased to 30-40 in 2016-2018, which is within the range of what is normally observed in the anoxic parts of the EGB (~20-70; Sommer et al., 2017; Viktorsson et al., 2013; Fig. 2b). These low DIC:DIP flux ratios are the result of preferential regeneration of P from organic matter, causing an elevated sedimentary DIP release in the absence of Fe oxides (Jilbert et al., 2011; Steenbergh et al., 2011, 2013), and release of DIP adsorbed to Fe oxides that are shuttled from shallower areas (Dellwig et al., 2010). The consistent DIC:DIP ratio in all years apart from 2015 (right after the MBI) implies that other factors than release of temporarily trapped $P_{Fe}$ contributed to the increased benthic DIP efflux at station F in 2016-2017."*

Line 231: It is not clear what is meant by "change over time in response to external factors". Please clarify.

We have modified the sentence.
*"The variation between replicate cores in 2016 raises the question whether changes in pore-water DIP, $P_{ex}$ and $P_{Fe}$ at station D were caused by spatial variability, or if there was a change over time in response to oxygenation."*

Line 233: I think that the SMHI data referred to here is presented in Fig. A1. If so please cite the figure.

A reference to Fig. A1 has been added.

Line 318-320: Should the units in the text be the same as those in Fig. 5 (mmol/m2/d)?

The numbers presented here are the total, time-integrated (3.13 years) releases of DIP in the different scenarios. The unit mol m$^{-2}$ (per 3.13 years) is thus correct, but we understand that it may cause confusion and have modified the text:

*"To estimate the impact on the water column P inventory in the EGB, we propose three scenarios for the total, time-integrated DIP release from the sediment between March 1 2015 and April 16 2018 (3.13 years, Fig. 5a)."*

The total DIP releases correspond to the areas underneath each of the three curves in Fig. 5a; we have split Fig. 5a into three panels and shaded the areas, and modified the figure caption, to clarify this.

**Reviewer 3**

Review of Hylen et al. Deep-water inflow event increases sedimentary phosphorus release on a multi-year scale

I found this paper to be clearly communicated and I appreciate the targeted focus of the study. I think the ability to capture a sediment-water flux response to a series of inflow events in the Baltic Sea is unique and the results are highly relevant to our understanding of the biogeochemical responses to eutrophication and oxygen depletion. The combination of in-situ sediment-water fluxes and porewater analysis (and supplemented by available monitoring) provides a sufficient dataset to explore the questions proposed by the authors. I do think some critical questions should be addressed in a revision of the paper, notably with respect to the impact of the seasonality of sediment-water fluxes, the ratio of DIC to DIP fluxes, and the balance between the inflows causing elevated organic deposition versus stimulating additional remineralization through oxygenation.

We thank the reviewer for constructive and valuable feedback.

I provide some specific and general comments below:

- I think Figure 1 is unnecessary. The main features of the feedbacks have been previously well described in the literature and the authors description of the feedback in the text is clear and adequate.

  We agree, the figure has been removed.

- Figure 2 – please specify the depths included to compute "bottom water" oxygen

  This information has been added to the figure caption.
  *"(c) Bottom-water oxygen ($O_2$) concentrations measured by optodes on the benthic chamber lander (average ± standard deviation) ~0.2 m above the sediment."*

- I think it could be stated in the paragraph in Line 120 that the prior fluxes (2008, 2010, and 2015) were performed using the same lander system as in the more recent fluxes.

  This information has been added.
  *"The fluxes from 2008, 2010 and 2015 (Hall et al., 2017; Nilsson et al., 2019; Viktorsson et al., 2013), which were measured using the same lander systems as in this study, have been re-evaluated to assure that the same evaluation method is used and thus the results are comparable."*

- I would like the authors to enhance the discussion around the differences in sampling season between the earlier and later sediment-water fluxes. This is an important feature of the interpretation of the flux enhancement over time. Annual cycles of sediment-water fluxes in many

temperate estuaries involve substantial seasonal variation. Is there a reason that a deep, typically anoxic basin should be different? Are the other datasets to cite from the literature? What about seasonal temperature effects? Could a reactive pool of organic material that may have accumulated during colder months be available by April, where if a spring bloom was important, perhaps this material would have been exhausted by late summer, when some of the other fluxes were measured?

*It is indeed important to exclude potential seasonal effects. We have split section 3.3 (Increased mineralisation after the inflow) into subsections to make the discussion about potential causes for the increased mineralization rates more structured. One of the subsections (3.3.1) focuses on seasonal effects, bioturbation and changes in primary production. As discussed in the manuscript, sedimentation of fresh phytoplankton material during spring and summer should have given the highest benthic fluxes in 2008, 2010 and 2015 as these samplings took place during early summer, late summer and early autumn. The highest fluxes at station F were instead measured in 2016-2017, during samplings that were conducted in April right before the spring bloom. Seasonal patterns in organic matter input are thus unlikely to have driven the fluxes. As the reviewer points out, temperature changes can affect the organic matter mineralization. However, the temperature of the Eastern Gotland Basin deep water is constant over annual cycles and does not follow seasonal patterns. The deep-water temperature did increase slightly due to the MBI, but since the benthic fluxes were similar in years when the bottom water temperature differed (2008 and 2010 vs 2018), temperature changes are unlikely to have been the main factor. We have added a paragraph about temperature effects:*

*"Alternatively, organic matter mineralisation rates is known to increase with temperature (Arnosti et al., 1998). The water temperature below the halocline in the EGB does not follow seasonal patterns (Fig. A6; SMHI, 2021); different sampling months (April vs August/September) should thus not have affected the benthic fluxes. The MBI increased the bottom water temperature by ~1°C, however (Fig. A6; Liblik et al., 2018; SMHI, 2021). Despite lower temperatures in 2008 and 2010 compared to 2018, the benthic fluxes were similar during these years (Fig. 2a). Temperature differences are thus unlikely to alone have driven the benthic flux pattern."*

- I appreciate that all of the flux rates (DIC, DIP, DSi, NH4) increased together in 2016-2017 at station F. But in reference to the Redfield ration, there were substantial excess DIP fluxes in 2016 to 2018. The authors consider this DIP excess but do not provide a clear explanation as I can see. The later discussion on elevated Mn oxide deposition might be relevant, as would Mn oxides scavenge P from the water-column preferentially? They also don't explore sufficiently why the DIC flux exceeds the DIN flux from a Redfield perspective, which perhaps is due to denitrification?

*The DIC:DIP flux ratios in the EGB are low (20-70) also under long-term anoxia. The DIC:DIP fluxes observed in this study was on average 30-40, so well within the normal range for the area. We have expanded the section about flux ratios to describe this, see reply to question about Fig. 3 by reviewer 2.*

*We have also clarified that the DIC:DIN flux ratios agree well with C:N ratios observed in the sediment surface and in sinking particles in the Eastern Gotland Basin (Cisternas-Novoa et al., 2019; Nilsson et al., 2021).*

*"At both stations D and F, the DIC:DIN flux ratios were around 10-12 (Fig. 2b) rather than the Redfield ratio of 6.6 (106:16). The flux ratios in this study agree well with carbon:nitrogen ratios at*

*the sediment surface and in sinking particles in the Eastern Gotland Basin (Cisternas-Novoa et al., 2019; Nilsson et al., 2021)."*

We measured sedimentary denitrification during our samplings, but since the manuscript describing those data is still in preparation, it cannot be cited according to Biogeosciences' guidelines. In the years when there was oxygen in the bottom water denitrification was indeed active, but at very low rates. During long-term anoxia (i.e. in 2008 and 2010), nitrate is depleted and denitrification ceases. We therefore conclude that denitrification is an unlikely explanation for the elevated DIC:DIN ratio.

- In your conclusions, I think it is important to make a clear distinction between the inflows causing elevated POM flux and the inflows providing oxygen to enhance remineralization. If the former is more important, it would suggest that the inflow enhanced POM flux to sediments is the reason for the elevated sediment-water fluxes, and the oxygenation is simply associated with the inflow/flux in time. If this were true, it would dampen the argument that oxygenation might actually cause a transient enhancement of the "Vicious cycle"

  See reply below.

- I was hoping the authors might come back to the idea of the "viscous cycle" in a discussion paragraph and clearly put their results in the context of that hypothesis. It has been shown that DIP concentration in bottom-waters is related to oxygen concentration in the Baltic – and this is cited as indirect evidence of the enhancement of sediment-water DIP flux under low oxygen. The supplementary data you provide on bottom-water DIP concentrations doesn't appear to show any enhancement of water-column DIP, even with the elevated rates you measured. Is this because the DIP-enhancement of sediment-water flux you measured only occurred in a limited area (although the up-scaling you did suggests it could be meaningful over larger scales). What if the April measurements are not representative of most of the year, as they are clearly higher than the previous years when measurements were made during summer? If the DIP flux enhancement is due to oxygenation, do your measurements suggest a timescale of this enhancement (1-2 years) given reoxygenation of an area? I realize this is a lot to digest, but I think a paragraph in the discussion that evaluates your results in the greater context of the cycle, citing the limitations of your measurements, would help the paper.

  We agree that the implications of DIP release from enhanced POM input compared to enhanced POM mineralization were not clearly discussed. Following a suggestion from reviewers 1 and 2, we have calculated the Porg inventories in the sediment, which is used together with the measured benthic fluxes to show that enhanced POM input is the most likely reason for the elevated fluxes. We have added a discussion about the implications of this result, see replies to question 8 by reviewer 1 and major comment by reviewer 2.

  The supplementary figure of DIP concentrations only shows the top 10 m of the water column. We have added another supplementary figure with deep-water concentrations of DIP, which increase over the years following the MBI.

---

## Author Response (AR2)

**Responses to minor corrections on**

**"Deep-water inflow event increases sedimentary phosphorus release on a multi-year scale"**

Dear professor Middelburg,

Thank you for the feedback on the manuscript. Please see our responses (in blue) below.

l. 39: water-column stratification

> This has been corrected.
> "The central parts of the Baltic Sea have been naturally oxygen-depleted for thousands of years due to a strong water water-column stratification, a long water residence time and limited deep-water renewal (Zillén et al., 2008)."

l. 192-194: you use the argument that standard deviations do not overlap and then the phrase … which suggests that the fluxes were indeed elevated after the inflow. Your arguments would be much stronger if you would use a proper analysis of variance (ANOVA) to test whether fluxes are different/higher. Such a ANOVA based difference would strengthen the conclusiveness.

> We agree. In the original version of the manuscript, we excluded "non-significant" fluxes which occasionally resulting in few replicates per station and year and made the data set unsuitable for an ANOVA test. Since all fluxes are now included in the data set, it is possible to conduct a proper statistical analysis. We have conducted an ANOVA followed by Student-Newman-Keuls tests, which indeed show that the fluxes vary with time, that they were highest in 2016 and that the fluxes were highest at station F. We have added a paragraph to the methods section describing the statistical tests; the results are described in supplementary tables A2 and A3 as well as throughout *Results and Discussion*.

l.236: perhaps rephrase in consistency with to consistent with to avoid misreading as inconsistency.

> We have made the suggested change.
> "The excess inventory of $P_{ex}$ and $P_{Fe}$ suggest that the net P retention at station F by the sampling occasion in 2016 was 4.6 – 6.4 mmol P m$^{-2}$ (Fig. 3b), consistent with a study from the eastern part of the EGB where 12 mmol P m$^{-2}$ was estimated to have been retained due to the MBI (Hermans et al., 2019b)."

l. 258: Try to avoid on the other hand if there is no on the one hand prior.

> We have made the suggested change.
> "The 2017 inflow had a smaller effect on the sediment geochemistry at station D…"

l. 293: do you mean were missed or were missing? Just to check.

> We do mean missed, since we believe that this could be an issue with the method.

Section 3.3.2: I believe you could have utilized the increase in DSi fluxes to argue for enhanced deposition of diatoms material. (No action on your side required; just an observation).

> We agree, we have modified a sentence in section 3.3.3 to make this point.

> "The elevated DSi fluxes observed here could also indicate that an increased input of organic matter (with which biogenic silica is associated, such as diatom material), rather than the presence of oxygen, was causing the change in fluxes."